# The *Drosophila* ZAD zinc finger protein Kipferl guides Rhino to piRNA clusters

Lisa Baumgartner[1,2], Dominik Handler[1], Sebastian Wolfgang Platzer[1], Changwei Yu[1], Peter Duchek[1], Julius Brennecke[1]*

[1]Institute of Molecular Biotechnology of the Austrian Academy of Sciences, Vienna BioCenter, Vienna, Austria; [2]Vienna BioCenter PhD Program, Doctoral School of the University of Vienna and Medical University of Vienna, Vienna, Austria

**Abstract** RNA interference systems depend on the synthesis of small RNA precursors whose sequences define the target spectrum of these silencing pathways. The *Drosophila* Heterochromatin Protein 1 (HP1) variant Rhino permits transcription of PIWI-interacting RNA (piRNA) precursors within transposon-rich heterochromatic loci in germline cells. Current models propose that Rhino's specific chromatin occupancy at piRNA source loci is determined by histone marks and maternally inherited piRNAs, but also imply the existence of other, undiscovered specificity cues. Here, we identify a member of the diverse family of zinc finger associated domain (ZAD)-$C_2H_2$ zinc finger proteins, Kipferl, as critical Rhino cofactor in ovaries. By binding to guanosine-rich DNA motifs and interacting with the Rhino chromodomain, Kipferl recruits Rhino to specific loci and stabilizes it on chromatin. In *kipferl* mutant flies, Rhino is lost from most of its target chromatin loci and instead accumulates on pericentromeric Satellite arrays, resulting in decreased levels of transposon targeting piRNAs and impaired fertility. Our findings reveal that DNA sequence, in addition to the H3K9me3 mark, determines the identity of piRNA source loci and provide insight into how Rhino might be caught in the crossfire of genetic conflicts.

## Editor's evaluation

This paper reports fundamental insights into host-transposon interactions, and more specifically how specific genomic loci may be elected for producing anti-transposon piRNAs in *Drosophila*. The authors provide here compelling evidence that Kipferl, a ZAD zinc-finger protein, helps guide Rhino to G-rich motifs present in piRNA-producing loci in the female germline, demonstrating for the first time the existence of sequence-specific factors in piRNA biogenesis. The findings are of broad interest to the fields of heterochromatin and transposon biology.

*For correspondence: julius.brennecke@imba.oeaw.ac.at

Competing interest: The authors declare that no competing interests exist.

## Introduction

Eukaryotic genomes are littered with transposable elements. By constantly aiming to increase their copy number, transposons pose a major threat to genomic integrity and must therefore be tightly controlled. An important transposon silencing mechanism in animals is the PIWI-interacting small RNA (piRNA) pathway (*Siomi et al., 2011*; *Czech et al., 2018*; *Ozata et al., 2019*, *Senti and Brennecke, 2010*). Its loss leads to transposon de-repression and mobilization, DNA damage, and sterility. At the center of the piRNA pathway are Argonaute proteins of the PIWI clade loaded with 23-30nt long piRNAs which enable sequence specific repression of transposons at the transcriptional and post-transcriptional levels. To control the highly diverse transposon sequence repertoire, an equally diverse set of piRNAs must be generated. Transposon targeting piRNAs are typically encoded in specialized genomic regions (*Brennecke et al., 2007*; *Houwing et al., 2007*; *Aravin et al., 2008*).

**eLife digest** The genes within our DNA encode the essentials of our body plan and how each task in the body is achieved. However, our genome also contains many repetitive regions of DNA that do not encode functional genes. Some of these regions are genetic parasites known as transposons that try to multiply and spread around the DNA of their host. To prevent transposon DNA from interfering with the way the body operates, humans and other animals have evolved elaborate defense mechanisms to identify transposons and prevent them from multiplying.

In one such mechanism, known as the piRNA pathway, the host makes small molecules known as piRNAs that have sequences complementary to those of transposons, and act as guides to silence the transposons. The instructions to make these piRNAs are stored in the form of transposon fragments in dedicated regions of host DNA called piRNA clusters. These clusters thereby act as genetic memory, allowing the host to recognize and silence specific transposons in other locations within the host's genome. In fruit flies, a protein called Rhino binds to piRNA clusters that are densely packed to allow piRNAs to be made. However, it remained unclear how Rhino is able to identify and bind to piRNA clusters, but not to other similarly densely packed regions of DNA.

Baumgartner et al. used a combination of genetic, genomic, and imaging approaches to study how Rhino finds its way in the fruit fly genome. They found that another protein called Kipferl interacts with Rhino and is required for Rhino to bind to nearly all piRNA clusters. Since Kipferl can by itself bind to the sequences that Rhino needs to find, the results suggest that Kipferl acts to recruit and initiate Rhino binding within densely packed piRNA clusters. Further experiments found that, in flies lacking Kipferl, Rhino binds to regions of DNA called Satellite repeats, hinting that these selfish sequences may compete for Rhino for their own benefit.

The finding that Kipferl and Rhino work together to define the memory system of the piRNA pathway strongly advances our understanding of how a sequence-specific defense system based on small RNAs can be established.

These so-called piRNA clusters are enriched in transposon sequences, range from a few to several hundred kilobases in length, and act as heritable and adaptive transposon sequence libraries for the piRNA pathway.

Through their transcription, piRNA clusters provide the essential small RNA precursors with sequence information antisense to active transposon transcripts. The definition of specific genomic loci as piRNA precursor sources is therefore of central importance to determine the pathway's target spectrum. In *Drosophila* gonads, two general types of piRNA clusters are distinguished based on their mode of transcription: Uni-strand clusters are transcribed on one genomic strand from canonical RNA polymerase II promoters, giving rise to long, single stranded, and poly-adenylated transcripts (***Brennecke et al., 2007***; ***Mohn et al., 2014***; ***Goriaux et al., 2014***). The second type are dual-strand piRNA clusters, which are transcribed on both genomic strands (***Brennecke et al., 2007***; ***Mohn et al., 2014***; ***Klattenhoff et al., 2009***; ***Zhang et al., 2014b***; ***Chen et al., 2016***; ***Andersen et al., 2017***). Dual-strand clusters are active in germline cells and are embedded within heterochromatin. They differ from canonical RNA polymerase II transcription units by a lack of defined promoters, and by suppression of splicing and cleavage- and polyadenylation signals.

The molecular identity of dual-strand piRNA clusters is conferred by Rhino, a germline-specific variant of the canonical Heterochromatin Protein 1 a (Su(var)2–5) (***Vermaak and Malik, 2009***; ***Klattenhoff et al., 2009***). While Su(var)2–5 mediates gene silencing and chromatin compaction, Rhino acts in an opposite manner: It enables the recruitment of several germline-specific proteins, which are required to engage the cellular gene expression machinery at piRNA clusters and enable the nuclear export of the emerging transcripts (***ElMaghraby et al., 2019***; ***Andersen et al., 2017***; ***Mohn et al., 2014***; ***Kneuss et al., 2019***; ***Zhang et al., 2014b***; ***Zhang et al., 2012a***; ***Zhang et al., 2018***; ***Hur et al., 2016***; ***Chen et al., 2016***). In *rhino* mutant flies, dual-strand piRNA clusters lose their transcriptional capacity and convert into canonical heterochromatin. Consequently, piRNA production collapses, transposons are de-repressed, and flies are sterile. Based on these data, Rhino is the defining feature of dual-strand piRNA clusters, and its specific deposition is at the very center of steering piRNA populations to target 'non-self' (transposons) but not 'self' (genic loci).

Despite an increasing understanding of how Rhino orchestrates piRNA cluster expression, little is known about how cells control the specific deposition of Rhino onto chromatin. Current models involve a role of maternally inherited Piwi-piRNA complexes in the specification of piRNA cluster identity through Piwi-dependent deposition of H3K9 methylation, which is postulated to define dual-strand clusters at the chromatin level (*Mohn et al., 2014*; *Le Thomas et al., 2014*; *Shpiz et al., 2014*; *Akkouche et al., 2017*). In this manner, Rhino can be recruited adaptively to a changing transposon insertion profile. Indeed, Rhino's N-terminal chromodomain displays specific affinity to H3K9me3, which is consistently present at piRNA clusters (*Mohn et al., 2014*; *Le Thomas et al., 2014*; *Yu et al., 2015*). However, a large fraction of the pericentromeric, H3K9me3 enriched heterochromatin is not or only weakly bound by Rhino. Since its discovery, a major open question has therefore been how Rhino's genomic binding profile is defined.

Two observations indicate that, besides H3K9me3, cellular and genomic context impact Rhino's binding pattern. First, while Rhino is expressed both in ovaries and testes, the identity of dual-strand piRNA clusters differs in the two gonads despite identical DNA sequence (*Mohn et al., 2014*; *Klattenhoff et al., 2009*; *Chen et al., 2021*). Moreover, the genomic Rhino profile is dynamic in testes where the X-chromosomal *AT-chX* cluster attracts most of the cellular Rhino pool in differentiating spermatocytes but not at earlier developmental stages (*Chen et al., 2021*). Second, while all stand-alone insertions of active transposons are silenced through piRNA-guided heterochromatin formation, only around 20% of the insertions of any given transposon are, by unknown mechanisms, bound by Rhino (*Mohn et al., 2014*; *Shpiz et al., 2014*; *Radion et al., 2019*, *Akulenko et al., 2018*). Based on these two observations, Rhino-domains must be defined through a combination of H3K9me3 with additional, unknown activities that bind chromatin via Piwi-independent mechanisms.

Here, we show that the zinc finger protein CG2678/Kipferl defines the majority of Rhino's chromatin binding pattern in ovaries. The combinatorial readout of Kipferl's sequence-specific DNA binding together with Rhino's affinity to the H3K9me3 mark underlies the selective recognition of heterochromatic loci as substrates for piRNA precursor transcription. Our findings also suggest that distinct chromatin loci might compete for the cellular Rhino pool, offering insight into why Rhino is a fast-evolving protein.

## Results

### The H3K9me3 mark alone cannot explain the large diversity of genomic Rhino domains

As a foundation for studying how Rhino's chromatin binding specificity is defined molecularly, and how variable it is in fly strains with different transposon insertion profiles, we determined Rhino's chromatin occupancy genome-wide in two commonly used laboratory strains ($w^{1118}$ and MTD-Gal4) and in *iso1*, the strain underlying the *D. melanogaster* reference genome (*Hoskins et al., 2015*). Chromatin immunoprecipitation experiments followed by next-generation sequencing (ChIP-seq) confirmed that Rhino binds mostly to extended domains that vary greatly in size and often lack clearly defined boundaries (*Mohn et al., 2014*). We therefore divided the genome into nonoverlapping genomic 1-kb tiles and calculated the Rhino enrichment per tile in each strain. This revealed that a substantial fraction of the genome (e.g. 3.9 Mbp corresponding to ~3% of the analyzable genome in the $w^{1118}$ strain) showed greater than 4-fold enrichment (p<0.036, Z-score >2.1) for Rhino over input.

To analyze Rhino's chromatin binding profile, we divided the genome into pericentromeric heterochromatin and the generally euchromatic chromosome arms (based on H3K9me3 and Su(var)2–5 ChIP-seq data) (*Figure 1A*). The three well-described Rhino-dependent dual-strand piRNA clusters – *cluster 38C*, *42AB*, and *80F* – were not assigned to either category but were analyzed separately as reference loci. Rhino enrichment at major dual-strand clusters but also within the recombination-poor, pericentromeric heterochromatin followed a highly similar pattern in all three fly strains, with only few loci displaying strain-specific quantitative differences (*Figure 1B* left, middle; *Figure 1—figure supplement 1A*). Many Rhino domains in pericentromeric heterochromatin (e.g. *Figure 1C* panel 1) resembled the large piRNA clusters like *cluster 80F* (*Figure 1C* panel 2): they exhibited strong H3K9me3 signal, gave rise to abundant piRNAs, and were enriched in diverse transposon sequences. Within chromosome arms, several strain-specific Rhino domains were apparent (*Figure 1B* right). Based on manual inspection, these corresponded to genomic loci flanking stand-alone transposon insertions

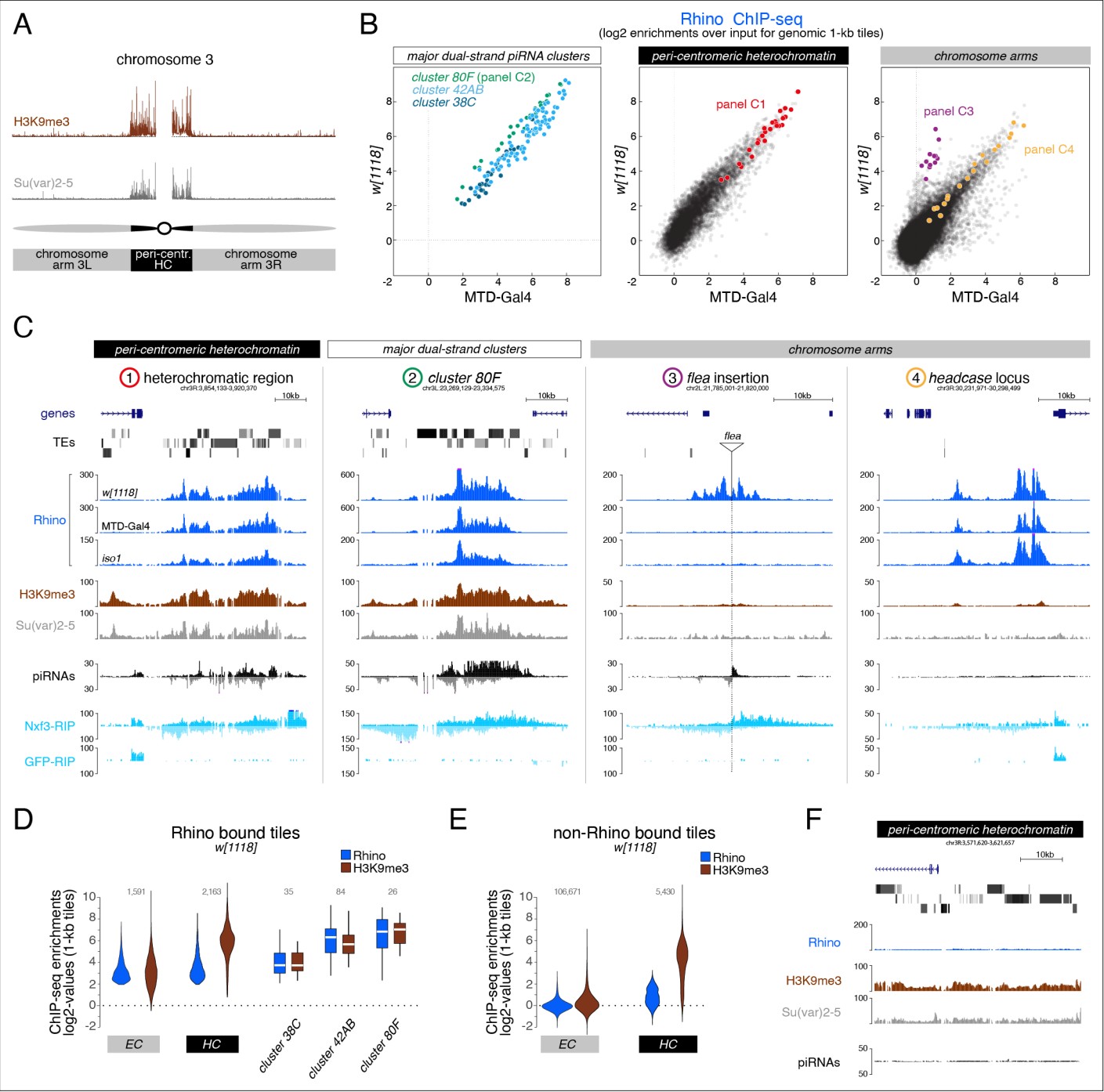

**Figure 1.** H3K9me3 is insufficient to define the large diversity of genomic Rhino domains. (**A**) ChIP-seq enrichment (genome unique reads; 1-kb tiles, one representative replicate each) of H3K9me3 and Su(var)2–5 along the assembled chromosome 3 sequence in $w^{1118}$ ovaries. Pericentromeric heterochromatin and euchromatic chromosome arms are indicated. (**B**) Scatter plot comparing average log2 ChIP-seq enrichments for Rhino in ovaries from $w^{1118}$ (n=2) versus MTD-Gal4 >$w$-sh (n=3) strains (1-kb tiles separated into pericentromeric heterochromatin and chromosomal arms; piRNA clusters *38C*, *42AB*, and *80F* are shown separately; colored 1-kb tiles correspond to example loci in panel C). (**C**) UCSC genome browser tracks depicting the diversity of Rhino domains. 1: heterochromatic, transposon-rich locus; 2: piRNA cluster *80F*; 3: strain-specific *flea* insertion (***ElMaghraby et al., 2019***) in $w^{1118}$; 4: Rhino domain proximal to euchromatic *headcase* locus. Unless indicated otherwise, data are from $w^{1118}$ovaries (ChIP and RIP signal are depicted as coverage per million reads, piRNA coverage normalized to miRNA reads, data is displayed for one representative replicate). GFP-RIP-seq serves as control for non-specific mRNA binding. (**D, E**) Violin plots showing average log2 fold enrichment of Rhino ChIP-seq over input for 1-kb tiles (n=2) from $w^{1118}$ ovaries. Tiles were grouped into Rhino-bound and non-Rhino-bound based on a cutoff of fourfold enrichment (corresponding to p=0.036, Z-score=2.1) of Rhino ChIP-seq signal over input in each replicate experiment. Rhino-dependent piRNA clusters *38C*, *42AB*, and *80F* were analyzed

*Figure 1 continued on next page*

*Figure 1 continued*

separately (shown as box plots due to low number of tiles). Box plots show median (center line), with interquartile range (box) and whiskers indicate 1.5x interquartile range. (**F**) UCSC genome browser tracks depicting a pericentromeric heterochromatin locus marked by H3K9me3 and bound by Su(var)2–5, but not Rhino (ChIP-seq signal: coverage per million reads; piRNA coverage normalized to miRNA reads; all data obtained from $w^{1118}$ ovaries, data is displayed for one representative replicate).

The online version of this article includes the following figure supplement(s) for figure 1:

**Figure supplement 1.** Relationships between methylated H3K9 and chromatin target sites of HP1 family members.

(e.g. a *flea* insertion found only in the $w^{1118}$ strain (*Figure 1C* panel 3)). The majority of euchromatic Rhino domains, however, were shared among all three strains (*Figure 1B* right; *Figure 1—figure supplement 1A*). Based on the *iso1* data and the reference genome, many of these shared domains were not associated with transposon insertions. Upstream of the *headcase* locus, for example, Rhino was strongly enriched in all strains, although this locus lacks transposon sequences and does not produce abundant piRNAs (*Figure 1C* panel 4). Nonetheless, these loci displayed dual-strand transcription based on RIP-seq data for Nxf3 (*ElMaghraby et al., 2019*), the piRNA precursor specific nuclear export factor, indicating that these loci were functional Rhino domains (*Figure 1C*). The poor piRNA output from these transposon-devoid loci is therefore not due to non-functional Rhino, but most likely due to a lack of piRNA target sites for Aub/Ago3-mediated transcript cleavage in the emerging transcripts, which greatly stimulate piRNA biogenesis (*Mohn et al., 2015*; *Han et al., 2015*). Altogether, our data demonstrate that Rhino, in a largely strain-independent manner, binds to thousands of remarkably diverse genomic loci that include, but are not limited to sources of dual-strand piRNAs.

Consistent with a reported role of H3K9 methylation in dual-strand piRNA cluster biology, Rhino domains in both pericentromeric heterochromatin and within chromosome arms consistently showed enrichment in H3K9me3 (*Figure 1D*). However, H3K9me3 levels did not directly correlate with Rhino enrichments: First, enrichments of H3K9me3 were considerably higher for Rhino domains within pericentromeric heterochromatin compared to those in chromosome arms, despite comparable Rhino enrichments (*Figure 1D*). Second, numerous loci in heterochromatin with high H3K9me3 signal were not or only poorly bound by Rhino (*Figure 1E and F*). Besides H3K9me3, the recombinant Rhino chromodomain also binds to H3K9me2, yet with lower affinity (*Mohn et al., 2014*; *Le Thomas et al., 2014*; *Yu et al., 2015*). As for H3K9me3, however, a large fraction of H3K9me2 was found outside of Rhino-bound loci (*Figure 1—figure supplement 1*). In contrast, Su(var)2–5 enrichments followed H3K9me2/3 levels at almost all genomic loci (*Figure 1—figure supplement 1C* compared to *Figure 1—figure supplement 1B*). This was also true for a tagged Su(var)2–5 protein expressed exclusively in germline cells where it competes with Rhino for binding to H3K9me3. Thus, methylation of H3K9 alone, while being a hallmark of Rhino domains, cannot explain Rhino's chromatin profile.

## The ZAD-zinc finger CG2678 interacts with Rhino and binds Rhino domains genome-wide

Sequence-specific DNA binding proteins can guide HP1 family proteins to chromatin. HP1b and HP1c, for example, depend on the zinc finger proteins Woc and Row for chromatin binding to thousands of sites that lack signal for Su(var)2–5 and even H3K9me2/3 (*Figure 1—figure supplement 1D*; *Font-Burgada et al., 2008*). Considering this, we hypothesized that undiscovered proteins with affinity to DNA motifs or to other histone marks collaborate with H3K9me3 to determine Rhino's genomic target loci.

To identify Rhino-interacting proteins, we performed a yeast two-hybrid screen using full length Rhino as bait and a cDNA library obtained from ovary mRNAs as prey. We identified 26 putative interactors among 175 sequenced colonies (*Supplementary file 2*), among them the known Rhino-interactor Deadlock (83 independent clones) (*Mohn et al., 2014*). With twelve independent clones, the uncharacterized protein CG2678 was the second most enriched screen hit. CG2678 further stood out among the other identified interactors because of its domain architecture: *CG2678* encodes a predicted DNA binder whose two annotated protein isoforms carry one or two arrays of $C_2H_2$ zinc fingers (ZnFs) (*Figure 2A*, *Figure 2—figure supplement 1*). Based on an N-terminal ZnF associated domain (ZAD), CG2678 is a member of the large group of ZAD-ZnF proteins that have radiated

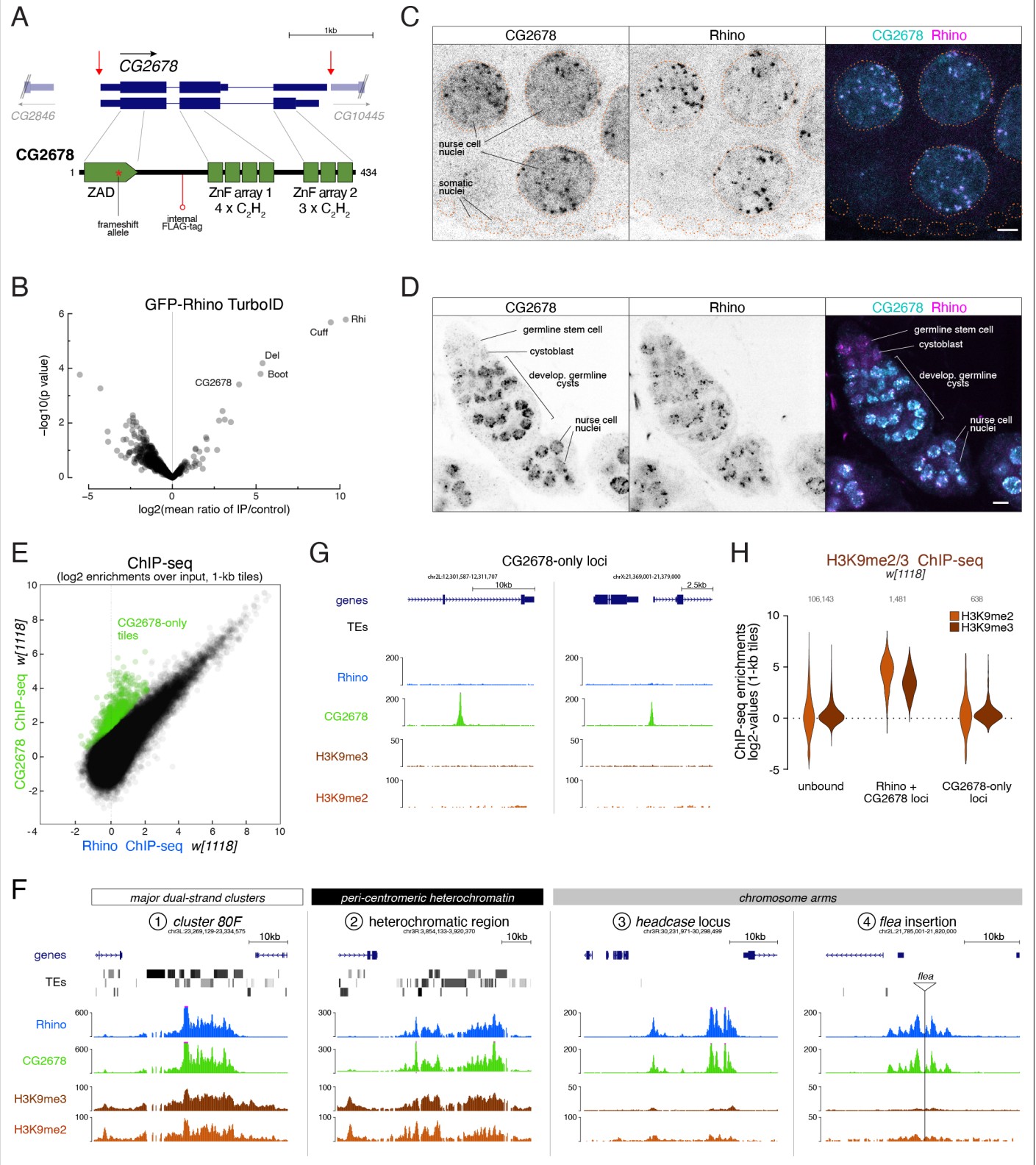

**Figure 2.** The ZAD-zinc finger CG2678 interacts with Rhino and binds Rhino domains genome-wide. (**A**) Genomic *CG2678* locus depicting the two annotated transcripts and CG2678 protein domain architecture (location of the frame shift mutation (red asterisk), internal 3xFLAG affinity tag (red circle), and cleavage sites for full locus deletion (red arrows) are indicated). (**B**) Volcano plot showing fold enrichment of proteins determined by quantitative mass spectrometry (*Doblmann et al., 2019*) in GFP-Rhino TurboID samples versus nuclear GFP TurboID control (n = 3 biological replicates;

*Figure 2 continued on next page*

*Figure 2 continued*

statistical significance based on two-sided t-test; P values corrected for multiple testing (Benjamini–Hochberg), see also *Figure 2—figure supplement 2A* and *Figure 2—source data 1*). (**C, D**) Confocal images of nurse cell nuclei (**C**) and germarium (**D**). Single channel and merged color images depict immunofluorescence signal for endogenous CG2678 (left, cyan) and Rhino (middle, magenta). Scale bar: 5 µm, dotted line: nuclear outline based on DAPI. (**E**) Scatter plot depicting correlation of log2-fold Rhino versus CG2678 ChIP-seq enrichment in $w^{1118}$ ovaries (average of 2 replicates each). CG2678-only tiles are highlighted in green and were defined by significantly higher enrichment for Rhino than CG2678 (n=2, Z-score=3), plus a Rhino enrichment of max. 4-fold in two independent experimental replicates. (**F, G**) UCSC browser tracks illustrating CG2678 signal at diverse Rhino domains (**F**) and at CG2678-only peaks (**G**). ChIP-seq signal is shown as coverage per million sequenced reads for $w^{1118}$ ovaries for one representative replicate. (**H**) Violin plots depicting log2-fold enrichment of H3K9me2 (orange, n=1) and H3K9me3 (brown, n=2) at euchromatic 1-kb tiles bound by neither Rhino nor CG2678, both proteins, or CG2678 only. Classification into groups was performed based on binary cutoffs for Rhino (fourfold) and a linear fit for CG2678 co-occupancy in two independent replicate ChIP-seq experiments from $w^{1118}$ ovaries to extract CG2678-only tiles highlighted in (**E**).

The online version of this article includes the following source data and figure supplement(s) for figure 2:

**Source data 1.** Imputed enrichment and p-values (n=3) for quantitative mass spectrometry data underlying *Figure 2B*.

**Figure supplement 1.** Multiple sequence alignment of selected reciprocal best CG2678 homologs generated with mafft v7.505 (*Rozewicki et al., 2016*) and visualized with esprit v3.0 (*Robert and Gouet, 2014*).

**Figure supplement 2.** CG2678 is expressed in ovaries but not in testes and binds Rhino domains on chromatin.

extensively in insects (*Chung et al., 2007*; *Chung et al., 2002*). Only a handful of the >90 *D. melanogaster* ZAD-ZnF proteins have been studied, and these are involved in transcription (*Bag et al., 2021*; *Harms et al., 2000*), genome organization (*Maksimenko et al., 2015*; *Sabirov et al., 2021*), and heterochromatin biology (*Kasinathan et al., 2020*). As a predicted DNA-binding protein and a putative Rhino interactor, CG2678 was an intriguing candidate for a potential Rhino specificity factor. To ask whether Rhino and CG2678 interacted also in vivo, we performed a proximity labeling experiment (*Branon et al., 2018*). We generated transgenic flies expressing GFP-tagged Rhino and low levels of the biotin TurboID-Ligase fused to a GFP nanobody, isolated biotinylated proteins under denaturing conditions from ovary lysate, and performed quantitative mass spectrometry. This revealed that CG2678 was enriched in the Rhino-GFP sample versus a control sample at similar levels as the piRNA cluster factors Deadlock, Cutoff, and Bootlegger (*Figure 2B*; *Figure 2—figure supplement 2A*). Together, these data implicated CG2678 as a promising candidate for further characterization.

Like for *rhino*, *CG2678* mRNA levels were detected primarily in ovaries (Flybase gene expression atlas, *Figure 2—figure supplement 2B*, *Larkin et al., 2021*). Immunofluorescence experiments revealed CG2678 expression specifically in ovarian germline cells, with no detectable signal above background in somatic cells which lack Rhino-defined piRNA clusters (*Figure 2C*). In germline nurse cells, CG2678 was enriched in nuclei and accumulated in numerous discrete foci in a pattern that was almost indistinguishable from that of a Rhino staining (*Figure 2C*). Expression of CG2678 in ovaries, however, was not uniform: Single cell RNA-seq data from adult ovaries indicated low mRNA levels in germline stem cells and increased levels in differentiating nurse cells (*Rust et al., 2020*). Consistent with this, CG2678 protein, while abundant in differentiating cysts and polyploid nurse cells, was barely detectable in germline stem cells and cystoblasts despite these cells expressing Rhino (*Figure 2D*). CG2678 was further not detectable in testes (*Figure 2—figure supplement 2B, C*). These data were intriguing given that the identity of Rhino-dependent dual-strand piRNA clusters differs, for unknown reasons, between males and females (*Chen et al., 2021*).

ChIP-seq experiments using anti-CG2678 antibody confirmed that the co-localization of CG2678 and Rhino in nuclear foci was due to both proteins binding the same chromatin sites, as CG2678 co-occupied Rhino domains genome-wide in all three wild-type strains ($w^{1118}$, MTD-Gal4, and *iso1* strains) (*Figure 2E*; *Figure 2—figure supplement 2D*, E). Closer inspection revealed that CG2678 and Rhino bind to chromatin in a virtually indistinguishable pattern, often occupying broad chromatin domains. This was true at major piRNA clusters (e.g. *cluster 80F*), Rhino domains in pericentromeric heterochromatin, stand-alone transposon insertions in chromosome arms as well as loci like *headcase* that lack transposon sequences and piRNA output (*Figure 2F*).

Notably, 647 genomic 1-kb tiles exhibited significantly higher enrichment for CG2678 than for Rhino (Z-score >3) and were not enriched for Rhino (CG2678-only loci; 15.4% of all Kipferl-bound tiles) (*Figure 2E*). At these loci, the CG2678 ChIP-seq signal formed narrow peaks (e.g. *Figure 2G*). 98.7% of the CG2678-only loci resided within chromosome arms, suggesting that lack of H3K9 methylation at these sites might prevent stable Rhino binding. Indeed, almost all CG2678-only loci were devoid of

H3K9 methylation (*Figure 2G and H*). Overall, our findings revealed a molecular relationship between Rhino and the potentially sequence-specific DNA binder CG2678.

## Rhino's chromatin occupancy changes dramatically in *CG2678/kipferl* mutants

To investigate the role of CG2678 in Rhino biology, we generated mutant fly lines carrying frameshift alleles or a complete deletion of the *CG2678* locus (*Figure 2A*). *CG2678* null mutant flies were viable and, in western blot analysis, did not express detectable CG2678 protein (*Figure 3A*). *CG2678* mutant females showed normal egg-laying rates but strong fertility defects (*Figure 3B*). Insertion of the *CG2678* genomic sequence with an internal FLAG-tag (*Figure 2A*) at the deleted locus restored fertility to wildtype levels (*Figure 3B*).

Loss of CG2678 had no impact on Rhino levels (*Figure 3A*) but resulted in pronounced changes in Rhino localization: In wildtype nurse cells, Rhino accumulated in many distinct foci throughout the nucleus. In *CG2678* mutants, almost all Rhino signal gathered in a few large, often continuous structures adjacent to the nuclear envelope (*Figure 3C*). Depletion of *CG2678* mRNA by germline-specific RNAi caused a similar phenotype (*Figure 3—figure supplement 1A, B*), and expression of a tagged wildtype CG2678 protein in *CG2678* mutants restored wildtype Rhino localization to small, dispersed foci (*Figure 3—figure supplement 1C*). The prominent Rhino accumulations in *CG2678* mutants were enriched in H3K9me3 and the Rhino co-factors Deadlock and Nxf3, indicating that they were genuine chromosomal Rhino domains (*Figure 3—figure supplement 1D*).

Rhino domains tend to localize at the nuclear periphery also in wildtype ovaries, resulting in a putative piRNA precursor export and biogenesis compartment continuous with cytoplasmic nuage (*Zhang et al., 2012a*). We hypothesized that the pulling force provided by the piRNA precursor export pathway centered on Nxf3 drags Rhino domains to nuclear pore complexes (*ElMaghraby et al., 2019*). Indeed, in *CG2678,nxf3* double mutant flies, Rhino still accumulated in few, large foci but these were no longer confined to the nuclear envelope (*Figure 3C*). We termed *CG2678* 'kipferl' because of the prominent enrichments of Rhino in crescent-shaped structures that reminded us of a famous Austrian pastry.

To explore whether Rhino's altered nuclear localization was linked to changes in its chromatin occupancy, we performed Rhino ChIP-seq experiments in *kipferl*-depleted ovaries. In the absence of Kipferl, Rhino's chromatin association was severely reduced at most genomic sites (*Figure 3D and E*). These included the major piRNA cluster *80F*, nearly all Rhino domains in pericentromeric heterochromatin, as well as stand-alone transposon insertions and euchromatic Rhino domains devoid of transposon insertions (*Figure 3E and F*). A similar phenotype was apparent in ovaries from flies carrying *kipferl* null mutant alleles (*Figure 3—figure supplement 1E*). Importantly, Rhino loss at these various genomic loci was not caused by reduced H3K9 methylation levels, as H3K9me2/3 levels remained high at Rhino-bound regions upon depletion of Kipferl (*Figure 3G*; *Figure 3—figure supplement 1F*).

Only 4.6% of all 1-kb tiles exhibiting a more than fourfold Rhino enrichment (n=2) in control flies remained Rhino-bound in *kipferl*-depleted ovaries. Among these were mainly tiles of piRNA *cluster 42AB* and, to a lesser extent, *cluster 38C* (*Figure 3D*). Fluorescence in situ hybridization (FISH) experiments confirmed that clusters *42AB* and *38C* are transcribed in *kipferl* mutants (*Figure 3—figure supplement 1G*). In line with only low residual Rhino binding especially at *cluster 38C*, we observe only weak overlap between GFP-Rhino and the RNA FISH signal for these piRNA clusters. Most importantly, however, their respective nuclear RNA-FISH signal did not co-localize with the strong, elongated Rhino accumulations at the nuclear envelope (*Figure 3—figure supplement 1G*). We therefore hypothesized that the prominent Rhino foci in *kipferl* mutants correspond to repetitive loci not assembled in the reference genome or not identifiable using genome-unique reads. Based on an analysis of all ChIP-seq reads, the average Rhino enrichment on transposon consensus sequences varied between two and 30-fold in wildtype ovaries, was non-detectable in ovaries depleted for Rhino, and was strongly reduced in the absence of Kipferl (*Figure 3H*). In contrast, several Satellite sequences, foremost the *Responder* and *1.688* family Satellites that give rise to Rhino-dependent piRNAs in wildtype ovaries and testes (*Chen et al., 2021*; *Wei et al., 2021*), maintained Rhino occupancy or accumulated even more Rhino than in wildtype ovaries (*Figure 3I*).

Consistent with elevated Rhino occupancy at both Satellites, we observed a corresponding increase in nascent transcripts as indicated by the enhanced RNA FISH signal in nurse cell nuclei of

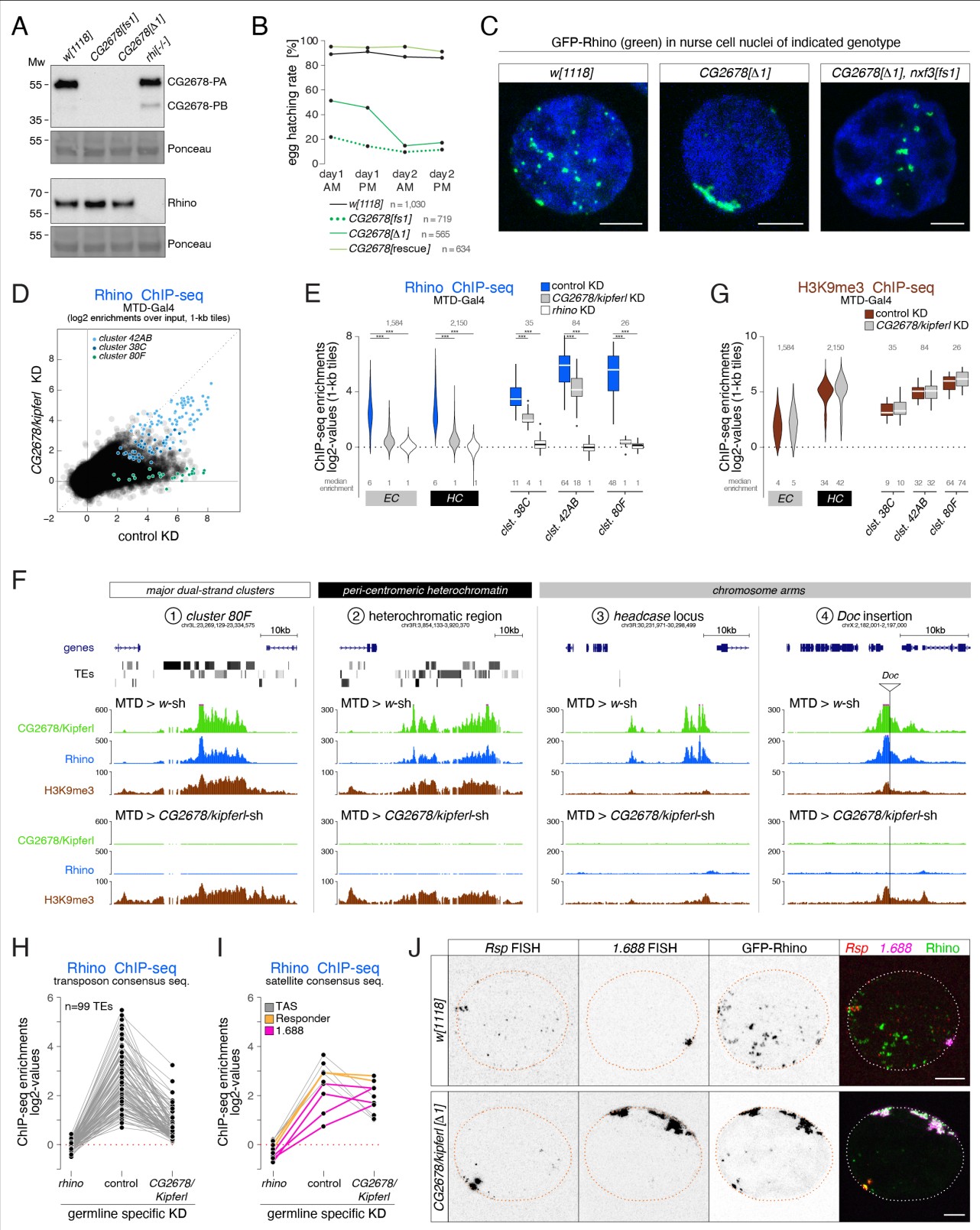

**Figure 3.** Rhino's chromatin occupancy changes dramatically in *CG2678/kipferl* mutants. (**A**) Western blot analysis verifying *CG2678* frame shift (*fs1*) and locus deletion (*Δ1*) alleles using a monoclonal antibody against CG2678 (top; CG2678-PB is a minor protein isoform) and depicting Rhino levels in the absence of CG2678 (bottom). Ponceau staining: loading control. (**B**) Time-resolved hatching rates for eggs laid by *w1118* control females in comparison to females carrying a *CG2678* frame shift (*fs1*), locus deletion (*Δ1*), or tagged rescue construct instead of the *CG2678* locus, respectively (AM, PM indicates

*Figure 3 continued on next page*

Figure 3 continued

egg laying time). Total number of eggs laid is indicated for each genotype. (**C**) Confocal images illustrating localization of GFP-Rhino in nurse cell nuclei of *w[1118]*, *CG2678* locus deletion (*Δ1*), and *CG2678,nxf3* double mutant females (scale bar: 5 µm). (**D**) Scatter plot of genomic 1-kb tiles contrasting average log2-fold Rhino ChIP-seq enrichment in ovaries with MTD-Gal4 driven *CG2678/kipferl* knock down versus control ovaries (average of two replicate experiments each). (**E**) Violin plots showing average log2-fold Rhino ChIP-seq enrichment in control (n=3) as well as *CG2678/kipferl* (n=2) or *rhino* (n=1) germline knockdown ovaries on Rhino-bound 1-kb tiles (defined in *Figure 1D*) in heterochromatin (HC) and chromosome arms (EC). piRNA clusters *38C*, *42AB*, and *80F* are depicted separately. *** corresponds to *P*<0,001 based on student's t-test. Box plots show median (center line), with interquartile range (box) and whiskers indicate 1.5x interquartile range. (**F**) UCSC browser tracks (ChIP-seq) depicting diverse Rhino domains in control and *CG2678/kipferl* germline knockdown ovaries (signal shown as coverage per million sequenced reads for one representative replicate). (**G**) Violin plots showing average log2-fold H3K9me3 ChIP-seq enrichment in control (n=3) and *CG2678/kipferl* (n=2) germline knockdown for Rhino-bound 1-kb tiles (defined in *Figure 1D*) in heterochromatin (HC) and along chromosome arms (EC). piRNA clusters *38C*, *42AB*, and *80F* are depicted separately. *** and n.s. corresponds to p<0.001 or p>0.05, respectively, based on student's t-test. Box plots show median (center line), with interquartile range (box) and whiskers indicate 1.5x interquartile range. (**H, I**) Jitter plots depicting the log2-fold Rhino ChIP-seq enrichments on transposon (**H**) and Satellite (**I**) consensus sequences in indicated genetic backgrounds. (**J**) Confocal images showing *Rsp* and *1.688* Satellite RNA FISH signal and GFP-Rhino in nurse cells of *w[1118]* or *CG2678/kipferl* mutant flies (scale bar: 5 µm).

The online version of this article includes the following figure supplement(s) for figure 3:

**Figure supplement 1.** CG2678/Kipferl impacts Rhino's nuclear localization and chromatin occupancy.

---

*kipferl* mutant ovaries compared to wildtype controls (*Figure 3J*). The FISH signal for *Rsp* and *1.688* transcripts in *kipferl* mutants overlapped precisely with the prominent Rhino accumulations at the nuclear envelope. *Rsp* and *1.688* Satellites form large repetitive tandem arrays in pericentromeric heterochromatin (*Wu et al., 1988*; *Abad et al., 2000*). The enormous size of these Satellite loci, particularly the *359 bp* array (*1.688* family) on the X-chromosome which extends over several mega-basepairs, likely explains why Rhino foci in *kipferl* mutants are so prominent (*Figure 3J*). The strong Rhino accumulations found in the center of nurse cell nuclei of *kipferl,nxf3* double mutant flies also overlapped with RNA FISH signal for *Rsp* and *1.688* Satellites, indicating that loss of *nxf3* did not alter Rhino's chromatin occupancy pattern (*Figure 3—figure supplement 1H*). In summary, Rhino's re-distribution from hundreds of genomic loci to the large, pericentromeric *Rsp* and *1.688* satellite arrays, which accumulate at the nuclear periphery in an Nxf3-dependent manner, explain the name-giving Rhino localization phenotype in *kipferl* mutants.

## *kipferl* mutant ovaries display piRNA losses and transposon de-repression

To investigate whether Rhino's altered chromatin occupancy in *kipferl* mutants affects piRNA production, we compared Argonaute-bound small RNA populations from *kipferl*-depleted ovaries to those from *rhino*-depleted and control ovaries. Total piRNA levels, normalized to miRNA reads, were reduced 4.5-fold in *rhino*-depleted but only 1.5-fold in *kipferl*-depleted ovaries (*Figure 4—figure supplement 1A*). We grouped genomic piRNA sources into somatic source loci (e.g. *flamenco*), Rhino-independent germline source loci (e.g. uni-strand *cluster 20*A) and Rhino-dependent germline source loci (e.g. dual-strand clusters *38C*, *42AB*, *80*F) (*Figure 4—figure supplement 1B*, *Mohn et al., 2014*). *kipferl*-depleted ovaries exhibited reduced piRNA levels only from Rhino-dependent germline source loci (*Figure 4A*). A prominent example was *cluster 80F*, where piRNA production collapsed in the absence of Kipferl to the same extent as seen in *rhino*-depleted ovaries (*Figure 4A and B*). At many other sites, piRNA losses were less severe compared to *rhino*-depleted ovaries. Among them were piRNA *clusters 38*C and *42AB*, consistent with residual Rhino binding to both clusters in *kipferl*-depleted ovaries (*Figure 4A and B*; *Figure 3D and E*).

The selective impact of *kipferl* depletion on germline-specific, Rhino-dependent piRNA source loci was also reflected when piRNAs mapping in antisense orientation to transposon consensus sequences were analyzed (*Figure 4C*). piRNAs targeting soma-controlled transposons (e.g. *Tabor*, *gypsy5*, *ZAM*; mostly originating from *flamenco*) were not affected in *rhino*- or *kipferl*-depleted ovaries. Among the transposons with dominating germline piRNA populations, nine are targeted by Rhino- and Kipferl-independent piRNAs originating from Rhino-independent source loci like *cluster 20A* (e.g. *297*, *roo*). All other transposons were targeted by Rhino-dependent piRNAs. For most of these elements, piRNA pools were reduced in *kipferl*-depleted ovaries (*Figure 4D* left, E). The few transposons that retained

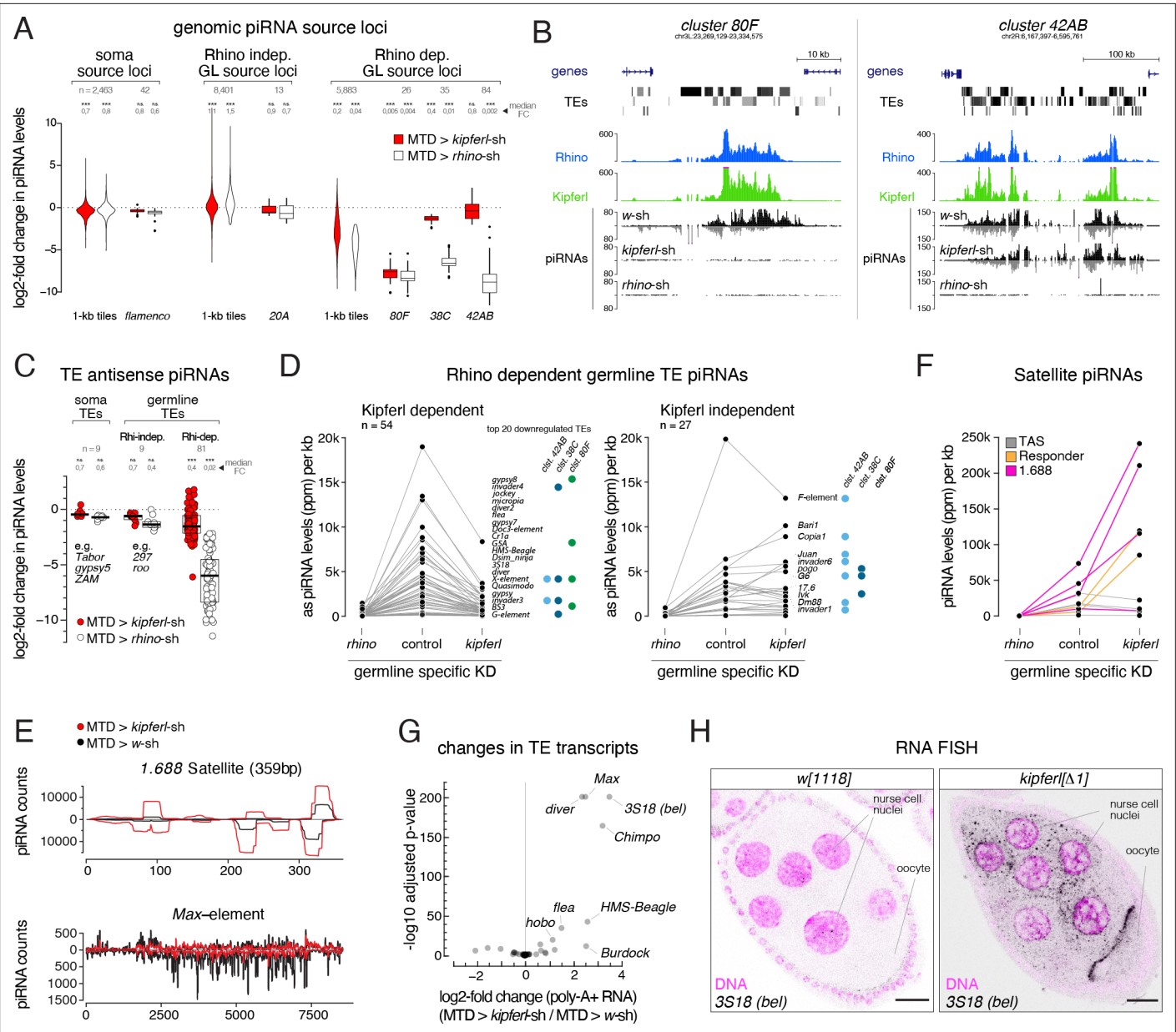

**Figure 4.** *kipferl* mutant ovaries display piRNA losses and transposon de-repression. (**A**) Violin plots showing log2-fold changes in levels of uniquely mapping piRNAs on 1-kb tiles relative to control upon MTD-Gal4 mediated knock down of *rhino* or *kipferl* (1-kb tiles were categorized into somatic source loci, Rhino-independent germline source loci and Rhino-dependent germline source loci according to *Figure 4—figure supplement 1B*, n=1). *** and n.s. corresponds to p<0.001 or p>0.05, respectively, based on Wilcoxon signed-rank test. Box plots show median (center line), with interquartile range (box) and whiskers indicate 1.5x interquartile range. (**B**) UCSC genome browser tracks displaying piRNA levels at clusters *80F* and *42AB* in control, *kipferl*, and *rhino* knock down ovaries (ChIP-seq signal from MTD-Gal4 control ovaries is depicted as coverage per million reads, piRNA coverage was normalized to miRNA reads, data is given for one replicate). (**C**) Jitter plot depicting log2-fold changes for piRNA levels mapping antisense to transposon consensus sequences in indicated MTD-Gal4 mediated knock downs compared to control (transposons classified analogous to panel A, n=1). *** and n.s. corresponds to p<0.001 or p>0.05, respectively, based on Wilcoxon signed-rank test. (**D, E**) Jitter plots showing piRNA levels (per kb sequence) in indicated genotypes mapping to transposons (antisense only) giving rise to Rhino-dependent piRNAs (**D**) or to Satellite repeats (**E**). Blue and green dots in panel D indicate fragments of the respective transposon in piRNA clusters *38C*, *42AB*, or *80F* (n=1, compare to *Figure 4—figure supplement 1C* for full mutant data, and see *Figure 4—source data 1*). (**F**) piRNA profiles across the consensus sequences for a representative transposon (*Max*) and the *359bp 1.688* Satellite. piRNA counts (normalized to miRNAs) are displayed for indicated genotypes. (**G**) Volcano plot depicting the log2-fold changes in poly-adenylated transposon transcripts in *kipferl*-depleted versus control ovaries (n=3). (**H**) Confocal images showing RNA FISH signal for *3S18* transcripts in *w[1118]* and *kipferl* null mutant ovaries (scale bar: 20 μm).

The online version of this article includes the following source data and figure supplement(s) for figure 4:

*Figure 4 continued on next page*

*Figure 4 continued*

**Source data 1.** Normalized small RNA-seq counts and their mappings to transposable element sequences.

**Figure supplement 1.** Loss of Kipferl affects the expression of repetitive elements but not mRNAs in ovaries.

piRNAs or even displayed increased piRNA levels had insertions in piRNA clusters *38C* or *42AB*, where Rhino binding was largely Kipferl-independent (*Figure 4D* right).

We finally analyzed the *TAS*, *Rsp*, and *1.688* Satellite repeats, which give rise to Rhino-dependent piRNAs (*Figure 4F*, *Chen et al., 2021*; *Wei et al., 2021*). While piRNA levels from sub-telomeric *TAS* Satellites remained unchanged, those derived from the *Rsp* and *1.688* Satellites increased strongly in *kipferl*-depleted ovaries compared to control, consistent with increased Rhino occupancy at these loci (*Figure 4E and F*; *Figure 3I*). As a result, Satellite piRNAs which accounted for only 4% of all piRNAs in wildtype ovaries, accounted for more than one in five piRNAs (21%) in *kipferl*-depleted ovaries. We observed a highly similar redistribution of the piRNA pool in *kipferl* null mutant ovaries (outliers were likely due to strain-specific differences in the transposon profile) (*Figure 4—figure supplement 1C*). Thus, in the absence of Kipferl, redistribution of Rhino from its natural binding sites to pericentromeric Satellite repeats results in a substantial loss of transposon targeting piRNAs and increased Satellite piRNAs.

The levels of poly-adenylated transcripts for a number of transposable elements increased significantly in *kipferl*-depleted ovaries (*Figure 4G*). RNA FISH experiments confirmed the de-repression of these elements in *kipferl* null mutant ovaries (*Figure 4H*; *Figure 4—figure supplement 1D*). In line with their upregulation, levels of piRNAs antisense to these elements were reduced in Kipferl-depleted ovaries. We note that several other elements with similar losses of antisense piRNAs were not derepressed. In general, transposon de-repression in ovaries lacking Kipferl was less severe compared to ovaries lacking Rhino (*Mohn et al., 2014*). This might be due to the milder loss of piRNAs in *kipferl* mutants compared to *rhino* mutants, or due to Kipferl being expressed only upon germ cell differentiation.

Our combined genetic data suggested that Kipferl acts specifically in the piRNA pathway. Consistent with this, loss of Kipferl had hardly any effect on overall gene expression, similar to the piRNA-specific factor Rhino (*Figure 4—figure supplement 1E*). By defining most of Rhino's genomic binding sites and by preventing aberrant Rhino accumulations at Satellite repeats, Kipferl enables nurse cells to mount a robust and diverse piRNA defense that is essential for the tight repression of transposable elements. Based on these findings, we set out to understand the molecular connections between Kipferl, Rhino, and DNA/chromatin.

## Kipferl nucleates Rhino domains within H3K9me2/3 loci and binds guanosine-rich DNA motifs

Considering the strong impact of Kipferl depletion on Rhino's chromatin occupancy, together with the $C_2H_2$ ZnF arrays contained in Kipferl, we hypothesized that Kipferl is a direct, sequence-specific DNA binder that functions upstream of Rhino and recruits and/or stabilizes Rhino on chromatin at its binding sites if they are located within an H3K9me2/3 domain. To test this, we investigated Kipferl's chromatin binding capacity in the absence of Rhino. In *rhino* mutant ovaries, Kipferl was distributed throughout nurse cell nuclei rather than being enriched in discrete nuclear foci as in wildtype ovaries (*Figure 5A*). ChIP-seq experiments revealed that Kipferl nevertheless remained bound at most genomic 1-kb tiles that it occupied in wildtype ovaries (*Figure 5B*). Kipferl enrichment levels, however, often differed between *rhino*-depleted and control ovaries, with a strong decrease observed at several regions, including the largely Kipferl-independent piRNA clusters *42AB* and *38C*.

To further investigate the relationship of Rhino and Kipferl at chromatin, we determined Kipferl-bound regions in wildtype ovaries and compared the chromatin occupancy of both proteins at these sites in each other's absence (*Figure 5C*). This revealed a strong interdependence of Rhino and Kipferl at regions they co-occupied in wildtype ovaries, which included most heterochromatic Kipferl binding sites, as well as roughly 60% of the euchromatic sites. In agreement with our previous results, Rhino's chromatin occupancy was highly dependent on Kipferl. Loss of Rhino had a more complex effect on Kipferl's binding pattern: while the Kipferl ChIP-seq signal was reduced at sites co-occupied by Rhino, residual signal was visible at many of these loci. Furthermore, Kipferl binding at sites lacking Rhino

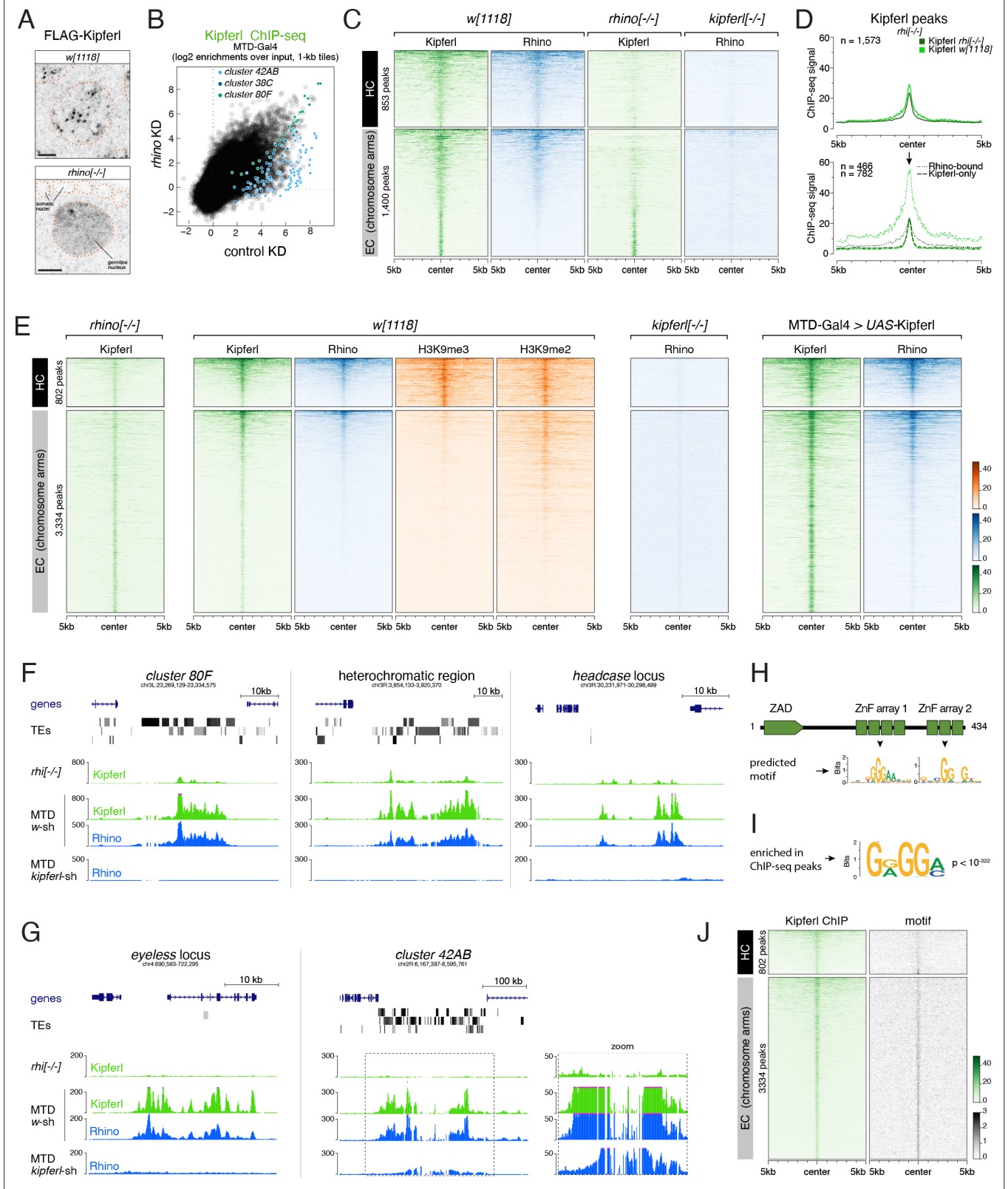

**Figure 5.** Kipferl binds guanosine-rich DNA motifs and nucleates Rhino domains within H3K9me2/3 loci. (**A**) Confocal images of nurse cell nuclei expressing FLAG-tagged Kipferl in indicated genetic backgrounds (scale bar: 5 μm; nuclear outlines based on DAPI as orange dotted line). (**B**) Scatter plot of genomic 1-kb tiles contrasting average log2-fold Kipferl ChIP-seq enrichment in ovaries with MTD-Gal4 driven *rhino* knock down (n=2) versus control ovaries (n=3). (**C**) Heat maps depicting ChIP-seq signal for Kipferl and Rhino in indicated genotypes from representative replicate experiments,

*Figure 5 continued on next page*

*Figure 5 continued*

centered on broad Kipferl peaks detected in two independent ChIP-seq experiments of $w^{1118}$ ovaries (data sorted by Kipferl signal in $w^{1118}$). (**D**) Meta profile showing Kipferl signal in wildtype and *rhino* mutant ovaries at all narrow Kipferl peaks determined in *rhino* mutant ovaries (top). The same peaks are divided into Rhino-Kipferl co-occupied (dotted line) and Kipferl-only peaks (dashed line) based on intersection with wildtype Rhino domains (bottom). (**E**) Heat maps depicting indicated ChIP-seq signal in indicated genotypes from representative replicate experiments centered on narrow Kipferl peaks detected in two independent ChIP-seq experiments of *rhino* mutant ovaries (data sorted by Rhino signal in $w^{1118}$). (**F, G**) UCSC genome browser tracks depicting Kipferl and Rhino ChIP-seq signals in indicated genotypes for diverse Rhino domains with (**F**) or without (**G**) pronounced Kipferl nucleation sites. Zoomed in view of *cluster 42AB* is given to appreciate small Kipferl enrichments in the peripheral part of the cluster (ChIP-seq signal depicted as coverage per million reads; piRNA coverage normalized to miRNA reads, data is given for one representative replicate). (**H**) Schematic representation of Kipferl protein domain architecture and binding motifs predicted for the two ZnF arrays (Princeton Cys2His2 PWM predictor). (**I**) Position weight matrix of Kipferl consensus motif determined by HOMER from top 3000 narrow Kipferl peaks found in two independent ChIP-seq replicates from *rhino* mutant ovaries. The GRGGN motif was found at least once in 83.4% of peaks versus 50.3% in control sequences ($P<10^{-322}$). (**J**) Heat map depicting Rhino-independent Kipferl ChIP-seq signal and GRGGN motif enrichment centered on narrow Kipferl peaks analogous to panel E, sorted by Kipferl signal in *rhi[-/-]* (motif count: # of motifs per non-overlapping genomic 100-bp window).

The online version of this article includes the following figure supplement(s) for figure 5:

**Figure supplement 1.** Kipferl binds guanosine-rich DNA motifs.

enrichment was fully maintained in the absence of Rhino, supporting Kipferl's ability to bind to chromatin without Rhino. Prompted by these observations, we determined Kipferl peaks in the absence of Rhino, refocusing our analysis on Kipferl's intrinsic, Rhino-independent chromatin binding sites. In *rhino* mutants, Kipferl was enriched at thousands of narrow peaks. Comparison of the Kipferl signal between wildtype and *rhino* mutants at these sites confirmed that Kipferl's chromatin occupancy was reduced at sites that were co-occupied by Rhino in wildtype ovaries, while it was unaffected at Kipferl-only peaks (*Figure 5D*). Moreover, while Kipferl bound to chromatin in similarly narrow peaks at stand-alone and Rhino-occupied regions in *rhino* mutants, Rhino seemingly supported the spreading of Kipferl to larger domains in wildtype ovaries.

Kipferl binding sites in *rhino* mutants were found both in pericentromeric heterochromatin (802 peaks) and within chromosomal arms (3334 peaks), and generally coincided with regions bound by Kipferl and Rhino in wildtype ovaries, demonstrating that Kipferl binds the same sites in both genotypes, yet in different patterns (*Figure 5E*). Kipferl's chromatin enrichment was strengthened and expanded from the narrow peaks seen in *rhino* mutant ovaries to broad domains in wildtype ovaries, specifically at peaks where Rhino was present. Kipferl peaks that did not show Rhino binding, on the other hand, did not widen into extended domains. Importantly, Rhino recruitment to Kipferl peaks depended on the presence of H3K9me2/3: Rhino occupied the majority of Kipferl peaks in pericentromeric heterochromatin where H3K9me2/3 is abundant. Within chromosomal arms, Rhino accumulated preferentially at those Kipferl sites that were within a local H3K9me2/3 domain. These findings supported a model in which Kipferl binds chromatin independently of Rhino, Kipferl binding sites act as Rhino nucleation sites when inside local heterochromatin, and Kipferl and Rhino cooperate and spread from nucleation sites into flanking heterochromatic regions, resulting in extended Rhino/Kipferl domains. Consistent with this, Rhino occupancy at and around Kipferl binding sites was dependent on Kipferl and overexpression of Kipferl in germline cells resulted in increased Kipferl binding at all sites, leading to the strengthening of Rhino/Kipferl domains within H3K9me2/3 domains, and the formation of additional small Rhino domains at pre-existing Kipferl sites with low, but detectable H3K9me2/3 levels (*Figure 5E*).

Kipferl's intrinsic chromatin binding profile in *rhino* mutants was strongly predictive of Rhino's chromatin occupancy, often mirroring the non-uniform enrichment of Rhino in wildtype ovaries, only at lower levels (*Figure 5F*). Indeed, 60% of Kipferl-dependent Rhino bound regions contained or were within 5 kb of a Kipferl nucleation site. However, at several loci (e.g. the *eyeless* gene), the extended Rhino domain in wildtype flies encompassed only very weak or no putative Kipferl nucleation sites (*Figure 5G*). Considering that Rhino depended on Kipferl also at these loci, we speculate that here, both proteins, supported by local H3K9-methylation and putative additional factors, are required for formation of a stable Rhino/Kipferl domain. Finally, in line with the largely Kipferl-independent Rhino occupancy at piRNA clusters *42AB* and *38C*, we find only weak intrinsic Kipferl binding at these loci (*Figure 5G*). Nevertheless, Kipferl bound strongly to these clusters in wildtype ovaries, implying that Rhino is capable of stabilizing Kipferl on chromatin. Upon closer inspection, Kipferl influenced Rhino's

chromatin profile even at the Kipferl-independent piRNA cluster *42AB*: While Rhino remained bound at the highly repetitive central regions of this cluster in the absence of Kipferl, several small Kipferl nucleation sites towards the periphery of the cluster seem to support the formation of an extended Rhino domain in wildtype ovaries (*Figure 5G*, zoom). Based on these data, we conclude that Kipferl is a major specificity factor for Rhino in ovaries, and that Kipferl cooperates with Rhino to form extended Rhino domains from defined nucleation sites, with both proteins supporting each other's chromatin occupancy. Our data further indicate that Kipferl is not the only Rhino specificity factor but demonstrate that it is also required for the stabilization of Rhino domains nucleated by alternative means.

According to the nucleation-site model, Kipferl is expected to bind to specific DNA motifs. Consistent with this, its two ZnF arrays are predicted to bind DNA with a specificity for guanosine-rich motifs (*Figure 5H*, *Persikov and Singh, 2014*). We determined sequence motifs that were enriched in Kipferl ChIP-seq peaks identified in *rhino* mutant ovaries. The top enriched motif (found in >80% of all peaks; $p < 10^{-322}$) closely matched the in silico predictions for Kipferl's ZnF arrays (*Figure 5I*). This GRGGN motif was locally enriched at experimentally determined Kipferl peaks, regardless of whether the peak was located in pericentromeric heterochromatin or chromosomal arms, or whether it was within a Rhino domain or constituted a Kipferl-only site (*Figure 5J*). Additional motifs detected at lower frequency and confidence often displayed variations of the same guanosine-rich motif (*Figure 5—figure supplement 1A*). In support of Kipferl's specificity for a GRGGN sequence, stable overexpression of FLAG-tagged Kipferl in cultured ovarian somatic stem cells (OSCs), which do not express Kipferl or Rhino, resulted in Kipferl binding at thousands of defined sites that were enriched in the GRGGN sequence motif (*Figure 5—figure supplement 1B*). Taken together, although future biochemical experiments will be required to confirm a direct interaction with DNA, our data support a model where the sequence-specific chromatin binding of Kipferl underlies the recruitment and/or stabilization of Rhino at chromatin.

## Structure-function analysis of Kipferl's DNA and Rhino binding activities

The yeast two-hybrid results (*Supplementary file 2*) and ChIP-seq analyses (*Figure 5*) suggested that Kipferl binds to both, Rhino and DNA. To examine how these two molecular activities are encoded within the Kipferl protein, we created flies that expressed truncated FLAG-tagged variants instead of endogenous Kipferl, lacking either one of the ZnF arrays (Kipferl$^{\Delta 1st\text{-}array}$, Kipferl$^{\Delta 2nd\text{-}array}$) or the N-terminal ZAD (Kipferl$^{\Delta ZAD}$) (*Figure 6A*). Kipferl$^{\Delta 1st\text{-}array}$ showed strongly reduced chromatin binding capabilities (*Figure 6B*). Deletion of the second ZnF array (Kipferl$^{\Delta 2nd\text{-}array}$) instead had only mild impacts. The N-terminal ZAD, which is characteristic for the >90 ZAD-ZnF family members in *D. melanogaster*, promotes anti-parallel homodimerization but has not been directly linked to DNA binding (*Jauch et al., 2003*; *Bonchuk et al., 2021*). Yeast two-hybrid experiments confirmed the dimerization capability of Kipferl's ZAD (*Figure 6—figure supplement 1A*). Deletion of the ZAD, and therefore abrogation of Kipferl dimerization, resulted in a global loss of Kipferl ChIP-seq signal (*Figure 6B*), consistent with studies about other ZAD-ZnF proteins (*Maksimenko et al., 2020*). Together, these findings indicated that Kipferl binds chromatin as a dimer, primarily via its first ZnF array.

To dissect the interaction of Kipferl with Rhino, we first examined the characteristic colocalization of Kipferl and Rhino in nurse cell nuclei. This revealed that both chromatin binding defective variants, Kipferl$^{\Delta 1st\text{-}array}$ and Kipferl$^{\Delta ZAD}$, failed to colocalize with Rhino (*Figure 6—figure supplement 1B*). In both genotypes, Rhino accumulated in prominent domains at the nuclear envelope, as seen in *kipferl* null mutant ovaries. Artificial dimerization of the Kipferl$^{\Delta ZAD}$ variant via the heterologous dimerization domain from the yeast Gcn4 transcription factor or the ZAD of Ouija board, a ZAD-ZnF protein not expressed in ovaries, restored co-localization of Kipferl with Rhino (*Figure 6—figure supplement 1B*, Kipferl$^{GCN4}$ and Kipferl$^{ouib}$). Thus, neither the ZAD, nor the second ZnF array are critical for binding to Rhino, suggesting that Kipferl's first ZnF array, besides its central role in chromatin binding, might also enable Rhino binding.

Yeast two-hybrid experiments, probing full length Rhino against Kipferl fragments, confirmed that Rhino interacts with Kipferl's first ZnF array, with no additional interaction interfaces being identified (*Figure 6C*, *Figure 6—figure supplement 1C*). To disentangle the putative DNA-binding and Rhino-binding activities of the first ZnF array, we further narrowed down the interaction between Rhino and Kipferl, revealing a critical role of the 4$^{th}$ ZnF (ZnF#4, *Figure 6C*; *Figure 6—figure supplement 1C,*

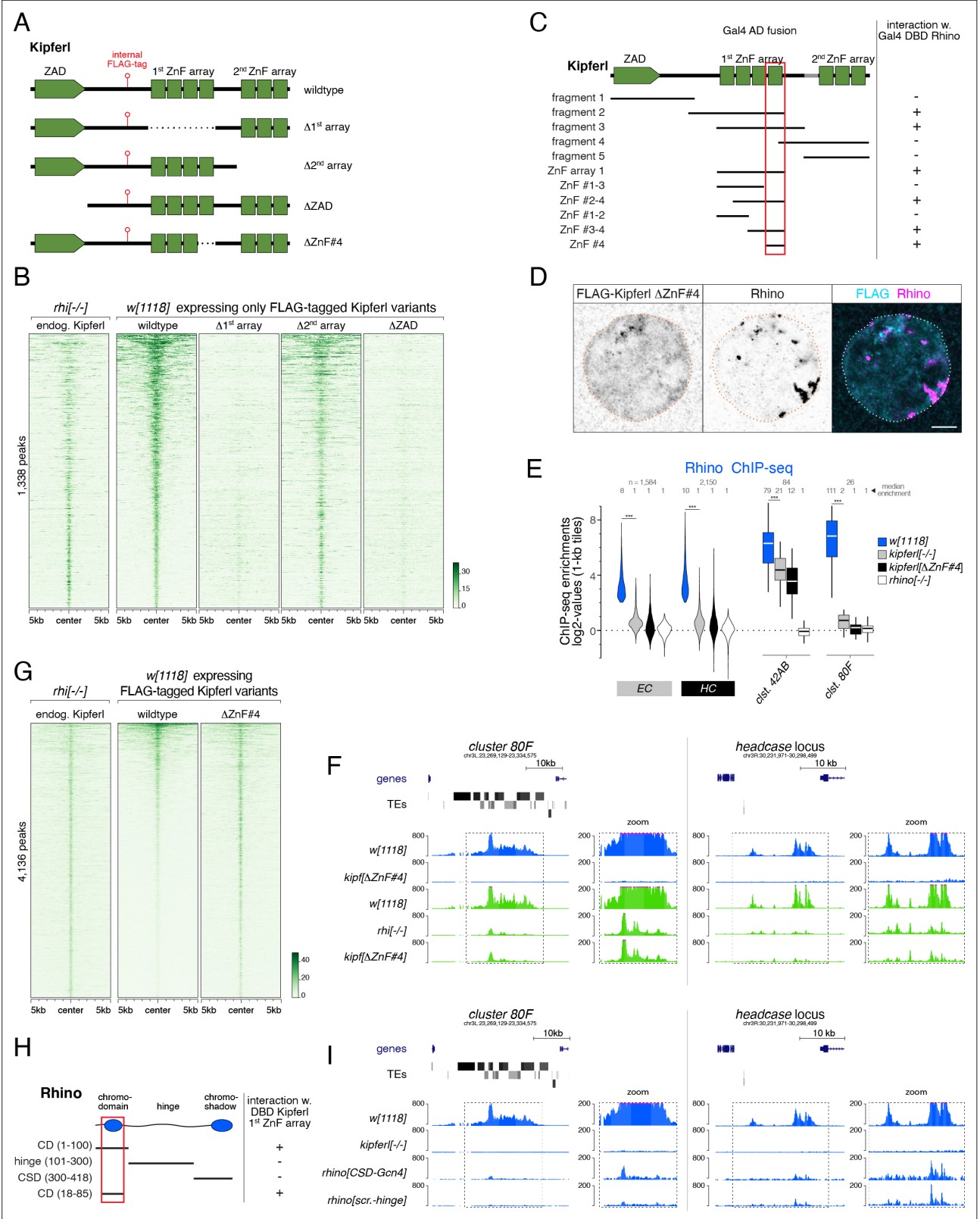

**Figure 6.** Structure-function analysis of Kipferl's DNA and Rhino binding activities. (**A**) Schematic representation of Kipferl rescue constructs harboring the wildtype protein sequence or indicated deletions, as well as an internal 3xFLAG tag. Rescue constructs were introduced into the endogenous *kipferl* locus via RMCE. (**B**) Heat map displaying Kipferl variant ChIP-seq signal centered on peaks bound by Kipferl in *rhino* mutants (data sorted by the ChIP signal detected for the wildtype Kipferl rescue construct; shown are only peaks that are Kipferl-bound in ovaries expressing wildtype tagged

*Figure 6 continued on next page*

*Figure 6 continued*

Kipferl). (**C**) Schematic overview of yeast two-hybrid experiments determining the minimal Rhino-interacting fragment in the Kipferl protein. Positive interactions between Kipferl fragments fused to the Gal4 activating domain (AD) and Rhino fused to the Gal4 DNA binding domain (DBD) are indicated. The minimal Kipferl fragment required for the Rhino interaction is highlighted in red (the grey bar indicates a 25 amino acid exon not contained in the reference genome which we identified in a subset of lab strains and included here for completeness). (**D**) Confocal images of a representative nurse cell nucleus depicting localization of Kipferl lacking ZnF#4 and Rhino in flies expressing only Kipferl$^{\Delta ZnF\#4}$ (scale bar: 5 μm). (**E**) Violin plots showing the average log2-fold enrichment of Rhino ChIP-seq signal over input for Rhino-bound 1-kb tiles (classified in *Figure 1D*) in *w$^{1118}$* (n=2), *kipferl* mutant (n=3), Kipferl$^{\Delta ZnF\#4}$ (n=1) and *rhino* mutant (n=1) ovaries. *** corresponds to p<0.001 based on student's t-test. Box plots show median (center line), with interquartile range (box) and whiskers indicate 1.5x interquartile range. (**F**) USCS genome browser tracks showing ChIP-seq signal (coverage per million sequenced reads for one representative replicate) for indicated proteins and genotypes at piRNA cluster *80F* (left) and the *headcase* locus (right). (**G**) Heat map showing indicated ChIP-seq signal, centered on narrow Kipferl peaks detected in two independent ChIP-seq experiments of *rhino* mutant ovaries (data sorted by ChIP-seq signal detected for wildtype Kipferl rescue). (**H**) Schematic overview of yeast two-hybrid experiments determining the minimal Kipferl-interacting fragment in Rhino. Positive interactions between Rhino fragments fused to the Gal4 activating domain (AD) and the first ZnF array of Kipferl fused to the Gal4 DNA binding domain (DBD) are indicated. The minimal Rhino fragment required for the Kipferl interaction is highlighted in red. (**I**) USCS genome browser tracks showing ChIP-seq signal (coverage per million sequenced reads for one replicate) for indicated Rhino variants at piRNA cluster *80F* (left) or the *headcase* locus (right).

The online version of this article includes the following figure supplement(s) for figure 6:

**Figure supplement 1.** Kipferl's 4th zinc finger is required and sufficient for the interaction with Rhino.

---

D). We tested this putative split-of-function mutant in vivo. In flies expressing Kipferl$^{\Delta ZnF\#4}$ instead of the endogenous protein, Rhino did not co-localize with Kipferl, and displayed the characteristic *kipferl* mutant phenotype (*Figure 6D*; *Figure 6—figure supplement 1B*). Moreover, Rhino lost its chromatin occupancy in *kipferl$^{\Delta ZnF\#4}$* flies in a pattern also seen in *kipferl* null mutants (*Figure 6E*). This included a complete loss of Rhino at Kipferl-dependent loci like *cluster 80F* or the *headcase* locus, but also remaining Rhino signal at Kipferl-independent loci like *cluster 42AB*, where Rhino's ChIP-seq signal in *kipferl$^{\Delta ZnF\#4}$* or *kipferl* null mutant flies was comparable (*Figure 6F*; *Figure 6—figure supplement 1E*). Kipferl$^{\Delta ZnF\#4}$ was diffusely localized in nurse cell nuclei akin to wildtype Kipferl in *rhino* mutants, and *kipferl$^{\Delta ZnF\#4}$* females were sub-fertile (*Figure 6D*; *Figure 6—figure supplement 1F*). Strikingly, Kipferl$^{\Delta ZnF\#4}$ retained full chromatin binding ability as it was enriched at Kipferl binding sites in a pattern closely mirroring that of wildtype Kipferl in a *rhino* mutant (*Figure 6G*). The Rhino- and chromatin-binding activities of Kipferl can therefore be uncoupled: ZnFs 1–3, supplemented by the second ZnF array, allow for putative sequence specific DNA binding, while ZnF 4 interacts with Rhino.

The finding that the 4$^{th}$ C$_2$H$_2$ ZnF fold in Kipferl is sufficient (*Figure 6C*) and required (*Figure 6D and E*) for Rhino binding was intriguing. Canonical HP1 interactors typically bind the chromoshadow domain dimer of HP1 proteins via PxVxL motif-containing peptides (*Thiru et al., 2004*). No such motif was found in Kipferl's 4$^{th}$ ZnF. We inverted the yeast two-hybrid assay to determine which region of Rhino interacts with Kipferl and discovered that Kipferl interacts with Rhino's chromodomain (*Figure 6H*; *Figure 6—figure supplement 1G*). Kipferl did not interact with the Rhino chromoshadow domain, the Rhino hinge region, or the chromodomain of Su(var)2–5. To test these findings in vivo, we generated flies that expressed Rhino variants with an artificial hinge or with a Gcn4 dimerization domain instead of the chromoshadow domain. Both Rhino$^{CSD-Gcn4}$ and Rhino$^{art.-hinge}$ failed to form extended Rhino domains like the wildtype protein, but were enriched at prominent Kipferl nucleation sites such as those in *cluster 80F* or upstream of the *headcase* gene (*Figure 6I*). Flies that expressed a Rhino variant with the chromodomain of Su(var)2–5 had rudimentary ovaries, precluding a meaningful ChIP-seq analysis, further supporting a model where the chromodomain contributes more than just binding to methylated H3K9. Our combined data indicate that Kipferl recruits Rhino to chromatin via a direct contact between its 4$^{th}$ ZnF and the Rhino chromodomain, and that functional Rhino is required for the extension and strengthening of Rhino/Kipferl domains.

## Kipferl is required for Rhino domains at diverse stand-alone transposon loci

The piRNA pathway's primary role is to silence transposable elements. Given their sequence diversity, it was surprising to find that a DNA binding protein with affinity for a short nucleotide motif is required for Rhino's chromatin occupancy and hence the determination of the ovarian piRNA pool. In fact, transposon sequences overall do not harbor more GRGGN motif occurrences than random genomic

tiles (*Figure 7—figure supplement 1A*). Instead, we observed a large spread of motif density among transposons (*Figure 7—figure supplement 1B*). The GC-rich *gypsy8* and *Rt1a/b/c* elements harbored between 29 and 38 motifs, while the AT-rich *Rsp* and *1.688* Satellites harbored none (*Figure 7A*; *Figure 7—figure supplement 1C*). Motif density per kilobase showed a significant correlation with Kipferl's intrinsic ability to bind to transposon sequences (as measured by Kipferl ChIP-seq enrichment in *rhino* mutant ovaries; $R=0.64$, $p<2.2e-16$) (*Figure 7B*). For most elements, the extent of Rhino-independent Kipferl enrichment was moreover directly correlated with their respective Rhino enrichment levels in wildtype ovaries (*Figure 7A and C*; $R=0.83$, $p<2.2e-16$). Exceptions were telomeric transposons and a few elements contained in clusters *42AB* or *38C*; these transposons were occupied by Rhino in wildtype ovaries despite no baseline Kipferl binding, and they maintained Rhino binding in *kipferl*-depleted ovaries, supporting the previous notion that Kipferl-independent chromatin recruitment mechanisms for Rhino likely exist (*Figure 7C*; *Figure 7—figure supplement 1D*). As the occurrence of the guanosine-rich Kipferl motif correlated with overall GC-content, enrichments for both Kipferl and Rhino on transposons also correlated with the elements GC-content (*Figure 7—figure supplement 1B, E, F*; $R_{motif-GC}=0.78$, $p<2.2e-16$; $R_{GC-Kipf}=0.71$, $p<2.2e-16$; $R_{GC-Rhi}=0.59$, $p<1.1e-13$). The correlation between GC-content and Rhino enrichment was abolished in ovaries depleted for Kipferl, indicating that Rhino's preferential binding to GC-rich transposons is due to Kipferl-mediated recruitment and/or stabilization of Rhino on DNA sequences with Kipferl motifs (*Figure 7—figure supplement 1G*, $R = 0.08$, $p=0.34$).

Many transposons did not show baseline Kipferl binding, yet their low-level Rhino enrichment in wildtype ovaries still depended on Kipferl. Moreover, several of these transposons (e.g. *Burdock*, *HMS-Beagle*, *3S18*) were strongly dependent on Rhino for their silencing, and were deregulated also upon loss of Kipferl, although to a weaker extent (*Figure 7A*; *Figure 7—figure supplement 1H, I*). The recent observation that flies mutant for the three piRNA clusters *38C*, *42AB*, and *20A* are fully fertile and do not show upregulation of these elements (*Gebert et al., 2021*) supports the proposal that stand-alone insertions likely act as independent Rhino domains (*Mohn et al., 2014*; *Shpiz et al., 2014*), providing piRNAs capable of targeting other insertions in trans. Considering that only about 20% of transposon insertions are transformed into Rhino domains (*Akulenko et al., 2018*), the average Rhino enrichment mapped to the consensus sequence would be low. We determined all stand-alone insertions of transposons lacking baseline Kipferl-binding in the MTD-Gal4 strain and displayed the levels of Rhino, Kipferl, H3K9me2/3, and piRNAs in the flanking genomic regions of these insertions (*Figure 7D*). While nearly all insertions were embedded in a local H3K9me2/3 domain, Rhino and Kipferl were only enriched at a subset of these insertions. RNAi-mediated depletion of Piwi or Kipferl resulted in loss of Rhino and piRNAs at these insertions. However, while Piwi loss impaired local heterochromatin formation, Kipferl loss did not (*Figure 7D*). We conclude that a combination of Kipferl and Piwi-dependent H3K9me2/3 is required to stabilize Rhino even on those transposons that do not contain strong Kipferl nucleation sites.

Based on manual inspection of individual transposon insertions in the different examined fly strains (illustrated in *Figure 7E*), we propose that Kipferl supports Rhino at stand-alone transposon insertions in one of two ways. In some instances, we find that Rhino-bound insertions occurred nearby a genomic Kipferl binding site, which might be a critical factor to establish a local Rhino domain. In other instances, we find no nearby Kipferl binding site. Here, Kipferl likely acts by stabilizing Rhino which might have been recruited through alternative nucleation factors akin to Kipferl's role at piRNA clusters *38C* and *42AB*, where Rhino remains chromatin bound in Kipferl mutant ovaries. Taken altogether, our data indicate that Kipferl acts as a recruitment factor for Rhino, and that it stabilizes the formation of extended Rhino domains, both at its own recruitment sites and at domains nucleated by alternative specificity factors.

## Discussion

This study provides direct evidence that DNA sequence is an important determinant of how germline cells define the chromatin binding sites of the fast-evolving HP1 variant protein Rhino. The discovery of Kipferl, a putative DNA-binding zinc finger protein, that cooperates with the H3K9me3 chromatin mark and serves as Rhino guidance factor critically advances our understanding of how dual-strand piRNA source loci are specified.

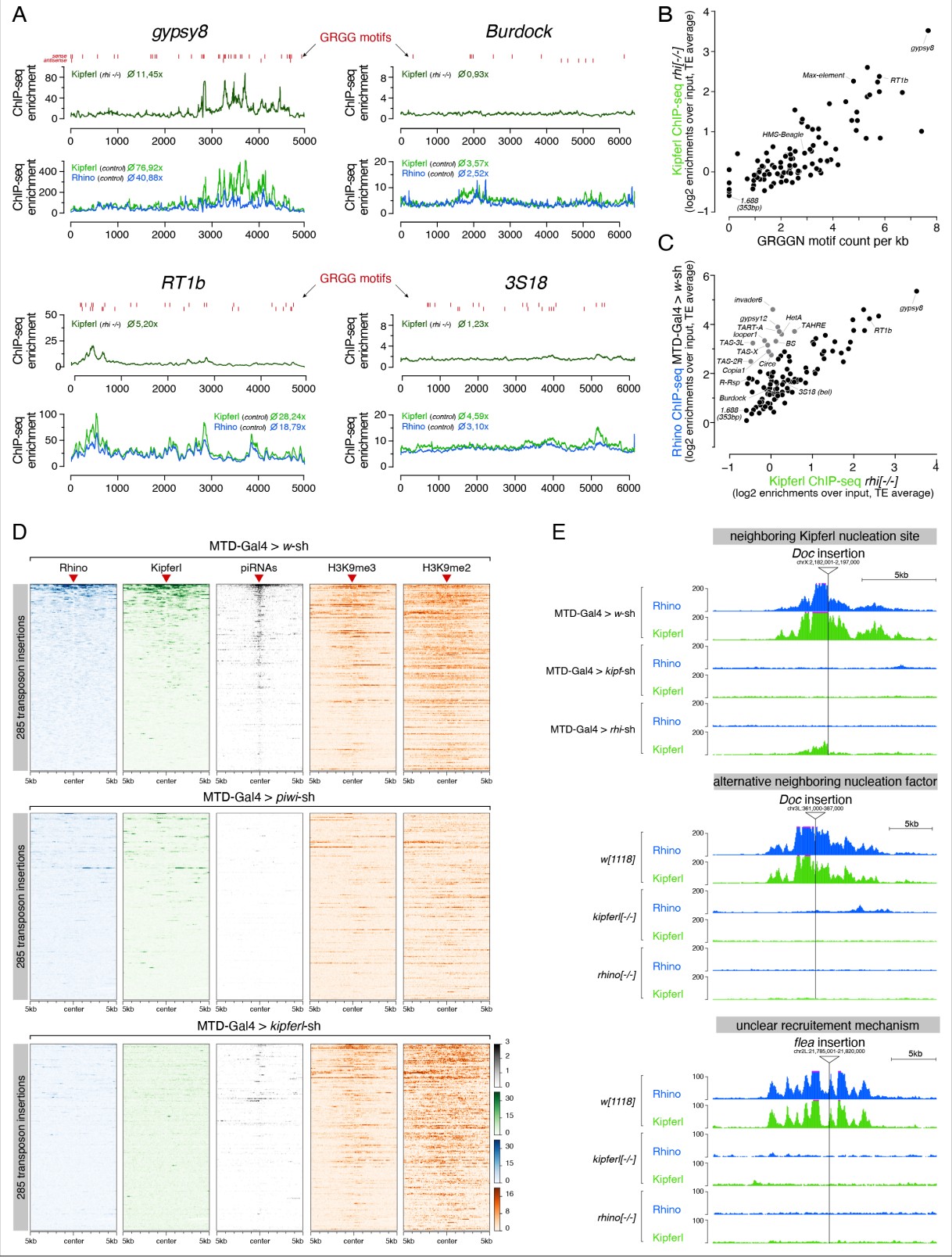

**Figure 7.** Kipferl is required for the establishment of Rhino domains at stand-alone transposon loci. (**A**) ChIP-seq enrichment profiles on consensus sequences of transposons with high (*gypsy8* and *Rt1b*) or low (*Burdock* and *3S18*) number of Kipferl DNA binding motifs per kb sequence. Indicated ChIP-seq signals are displayed as average enrichment over input in two (Kipferl) or three replicates (Rhino) of ovaries from *rhino* mutant (top tracks) or MTD-Gal4 >*w*sh control ovaries (bottom tracks). Red bars indicate motif instances on the sense or antisense strand. Numbers indicate average

*Figure 7 continued on next page*

*Figure 7 continued*

enrichment across the entire element. (**B**) Scatter plot correlating the GRGGN motif count (normalized to element length) to the Rhino-independent Kipferl ChIP-seq enrichment for each transposon (ChIP-seq enrichments depict average of two independent experiments). (**C**) Scatter plot depicting the relation between wildtype Rhino ChIP-seq enrichments and Rhino-independent Kipferl ChIP-seq enrichments per transposon (average of two (Kipferl) or three (Rhino) independent experiments; elements indicated in grey are bound by Rhino in a largely Kipferl-independent manner, see also *Figure 7— figure supplement 1D*). (**D**) Heat maps depicting indicated ChIP-seq signal in the genomic regions flanking 285 euchromatic stand-alone insertions (red triangles) of Rhino-dependent transposons with low Rhino-independent Kipferl binding (data sorted by the ChIP-seq signal detected for Rhino in MTD-Gal4 >*w*-sh ovaries). (**E**) UCSC genome browser tracks of stand-alone transposon insertions found in MTD-Gal4 or *w^1118^* strains, depicting examples of different potential modes of Kipferl dependency. ChIP-seq signal is shown as coverage per million reads for one representative replicate.

The online version of this article includes the following figure supplement(s) for figure 7:

**Figure supplement 1.** GC content and number of GRGGN motifs are predictive of Kipferl and Rhino binding at transposon loci.

By systematically comparing different *Drosophila melanogaster* strains, we find that Rhino domains in ovaries are largely identical between strains yet highly diverse in their genomic location, underlying sequence content, and piRNA output. Given the affinity of Rhino to the histone mark H3K9me3 (*Mohn et al., 2014*; *Le Thomas et al., 2014*; *Yu et al., 2015*), current models postulate that maternally inherited piRNAs define the cellular Rhino profile through epigenetic mechanisms during early embryogenesis (*Mohn et al., 2014*; *Le Thomas et al., 2014*; *Shpiz et al., 2014*; *Akkouche et al., 2017*). Indeed, Rhino domains are invariably accompanied by local di- and/or tri-methylation of H3K9. However, despite being targeted by maternally inherited piRNAs, not all transposon insertions are converted into a Rhino domain (*Akulenko et al., 2018*). Moreover, while Piwi depletion leads to a loss of Rhino from chromatin at stand-alone transposon insertions, where H3K9me2/3 is dependent on Piwi, most piRNA source loci, including the major piRNA clusters, retain Rhino binding in the absence of Piwi (*Mohn et al., 2014*). Finally, we discovered several transposon-free Rhino domains that produce only trace amounts of piRNAs. These domains are difficult to explain by maternally deposited piRNAs as Rhino specifiers, suggesting that additional, locus-specific mechanisms must be in place to define Rhino's chromatin binding profile. With Kipferl we present a factor capable of chromatin binding upstream of Rhino that has a major impact on nearly all Rhino domains in ovaries, irrespective of their position, transposon content, or piRNA output. Importantly, Kipferl does not influence the H3K9 methylation status, but rather stabilizes Rhino at most of its genomic targets, where H3K9me2/3 is provided through parallel pathways. In this manner, Rhino domains could be established at loci where heterochromatin coincides with Kipferl's intrinsic chromatin binding pattern. This model provides a possible explanation for the relatively low fraction of transposon insertions that are bound by Rhino, despite piRNA targeting at many other sites. Thereby, Kipferl, with its specific chromatin binding activity, serves as the first example of the long sought-after guidance cue(s) required for specifying Rhino's binding profile within heterochromatin.

While Kipferl binds to chromatin upstream of Rhino at a multitude of pronounced Rhino domains, not all Kipferl-dependent Rhino domains contain obvious Kipferl nucleation sites. Here, a combination of low intrinsic affinities of Kipferl and Rhino might underlie their observed co-dependence. Alternatively, additional guidance factors might exist, able to facilitate the recruitment, but not the stable binding of Rhino at these sites. The rare, Kipferl-independent Rhino binding at certain loci (e.g. clusters *42AB* and *38C*) argues in favor of additional specificity factors. An intriguing observation from our work is that Kipferl's spatio-temporal expression profile differs from that of Rhino: Kipferl levels in ovarian germline stem cells and cystoblasts are very low, and the protein is absent in testes. Kipferl's ovary-specific expression is therefore a likely contributor for the pronounced differences in Rhino landscapes between ovaries and testes (*Chen et al., 2021*). At the same time, it implies the existence of additional Kipferl-like factors or complementary molecular mechanisms that collaborate with H3K9me2/3 in testes or during early oogenesis where Rhino is functional in the absence of Kipferl. It will be interesting to see whether other Rhino guidance factors share Kipferl's function as Rhino stabilizer, or if Kipferl presents a special example of a Rhino guidance and support factor.

Our data indicate that Rhino binds to Kipferl via its chromodomain. Binding of client proteins has so far been assigned to the dimeric C-terminal chromoshadow domain of HP1 family proteins. By recruiting and/or stabilizing Rhino on chromatin via the chromodomain, Kipferl-binding would be compatible with the recruitment of downstream factors like Deadlock via Rhino's chromoshadow domain (*Yu et al., 2018*). Our genetic data imply that DNA binding of Kipferl as well as H3K9me2/3 binding of

Rhino are required for the formation of a stable Kipferl-Rhino complex on chromatin. Biochemical and structural work will be required to understand how the Kipferl-Rhino interaction evolved to be specific to the Rhino chromodomain and how it is compatible with simultaneous H3K9me2/3 binding. We were unable to generate Kipferl as a recombinant protein to study its DNA-binding potential or the Rhino-Kipferl interaction in vitro. It will further be interesting to investigate whether other chromatin readers utilize a similar mode of specialized recruitment for additional specificity. Known examples where ZnF proteins are involved in HP1 recruitment are the *Drosophila* HP1 variants HP1b and c, who rely on their cofactors Woc and Row for chromatin binding (**Font-Burgada et al., 2008**). In mouse embryonic stem cells, HP1beta/gamma bind to selected chromatin sites in a complex with the chromatin remodeler CHD4 and the activity-dependent neuroprotective protein (ADNP) (**Ostapcuk et al., 2018**). In both cases, interaction with ZnF proteins allows the H3K9me3-independent recruitment of HP1 variants to sites where their function has not been fully elucidated. This mode of recruitment is distinct from Kipferl's mode of action, which synergizes with underlying H3K9me2/3 at its binding sites to recruit Rhino to chromatin.

With an N-terminal ZAD, Kipferl belongs to the largest group of ZnF proteins in insects, with 92 members in *Drosophila melanogaster* (**Chung et al., 2007**; **Chung et al., 2002**). The functions of the few characterized ZAD-ZnF proteins are diverse and range from transcriptional regulation to chromatin organization and heterochromatin biology. Many ZAD proteins are preferentially expressed in ovaries and during early embryogenesis (**Shapiro-Kulnane et al., 2021**), and some of them, despite having essential functions, are fast evolving (**Kasinathan et al., 2020**). It is therefore possible that additional ZAD-ZnF proteins function as guidance factors for Rhino or other HP1 proteins in the germline. If that were the case, the transposon-genome arms race might be a key driver of the diversification of ZAD-ZnF proteins. Diversification of ZnF containing genes is also observed for KRAB or SCAN domain zinc fingers, whose radiation in tetrapods is believed to be fueled by the transposon conflict (**Bruno et al., 2019**). While KRAB-ZnFs act directly as repressors of transposon expression and show fast evolution of their DNA binding specificity, signatures of positive selection are not concentrated on ZnF domains in ZAD proteins (**Kasinathan et al., 2020**), indicating different evolutionary pressures acting on the two families of ZnF proteins.

At this point, we can only speculate about the processes that forced the evolution of a factor like Kipferl. The phenotype of *kipferl* mutants, however, holds important clues. In the absence of Kipferl, the piRNA profile in ovaries is strongly distorted. Most transposons exhibit reduced antisense piRNA levels, and for some this results in their de-repression and accumulation of transposon transcripts in the developing oocyte, potentially causative of the fertility defects of *kipferl* mutant females. One model would therefore be that Kipferl evolved as a dedicated Rhino specificity factor that targets the DNA sequence of some transposons, active in differentiating ovarian germline cells. While only few transposons contain strong Kipferl nucleation sites, we find that the strongest Kipferl binding sites genome-wide are within sequence fragments of the *gypsy8* and *DMRT1* family transposons. This indicates that the DNA motif recognized by Kipferl possibly originated from an ancient invasion of these now inactive elements, offering a potential evolutionary requirement for Kipferl as a direct recruitment factor for Rhino. The Rhino decoration of transposons lacking Kipferl nucleation sites might rely on sporadic genomic Kipferl binding sites which allow their specification as piRNA source loci as local chromatin state permits. The major piRNA cluster *80F* would be a prominent example, where fragments of *gypsy8* and *Rt1A* and *B* insertions induce the Kipferl-dependent recruitment of Rhino and subsequent piRNA production also from neighboring transposons. In support of this, the entire *cluster 80F* is not a strong piRNA source in testes, where Kipferl is not expressed (**Chen et al., 2021**). Thus, an initially specific homing of Rhino to a subset of transposons via Kipferl might have evolved into a network of transposon insertions within heterochromatin and at selected sites in euchromatic chromosomal arms that co-depend on Kipferl. This system would offer the potential for robust Rhino recruitment, at the cost of occasional off-target Rhino domains, when Kipferl motifs and H3K9 methylation coincide at regions lacking transposon sequences.

We can, however, also envision an alternative scenario, based on the second major phenotype in *kipferl* mutants: The dramatic re-localization of Rhino from hundreds of domains to Satellite arrays located within pericentromeric heterochromatin. Intriguingly, both the *Rsp* and the *1.688* Satellites are involved in genetic conflicts (**Larracuente and Presgraves, 2012**; **Ferree and Barbash, 2009**; **Chen et al., 2021**; **Ferree and Prasad, 2012**). The largest *1.688* Satellite array, the X-chromosomal

359 bp repeat, spans more than 10 Mbp on the pericentromeric X-chromosome and acts as a hybrid lethality locus in crosses between *D. melanogaster* and *D. simulans* (*Ferree and Barbash, 2009*; *Chen et al., 2021*). Females lacking this repeat, when mated to wildtype males, generate non-viable offspring while no defects occur in the reciprocal cross. This phenotype might stem from a requirement for maternally deposited small RNAs (siRNAs or piRNAs) to prevent mitotic catastrophe caused by uncontrolled *1.688* repeats (*Usakin et al., 2007*). The *Rsp* Satellite has been identified genetically as part of the Segregation Distorter system, a meiotic drive system in males (*Larracuente and Presgraves, 2012*). Both *Rsp* and *1.688* Satellites give rise to abundant piRNAs in a Rhino-dependent manner in testes and ovaries (*Wei et al., 2021*; *Chen et al., 2021*). If piRNA production from *Rsp* and *1.688* Satellites is important to maintain control of *1.688* repeats or to suppress the *Segregation distorter* locus during spermatogenesis, Satellites might have evolved to take advantage of the Rhino system and force piRNA production from their own loci. In this scenario, different genomic loci would compete for the cellular Rhino pool to recruit the Rhino-dependent transcription and export machinery. It is conceivable that proteins working in an analogous way to Kipferl are present at Satellites to sequester Rhino to these repeats. Intriguingly, Rhino is among the fastest evolving proteins in the fly genome (*Vermaak et al., 2005*). Previous models have postulated that the positive selection in Rhino is a consequence of the transposon-genome arms race (*Yu et al., 2018*). Kipferl is conserved among *Drosophila* species with stronger divergence outside the *melanogaster* clade. We do not find pronounced amino acid changes in Kipferl's zinc fingers involved in DNA binding or interaction with Rhino, nor in the ZAD, although the lack of amino acid polymorphisms at the *kipferl* locus precludes a conclusive analysis of selective forces acting on the protein (see also *Figure 2—figure supplement 1*). Nevertheless, our findings point to the provocative possibility that selfish Satellite sequences might be among the central drivers behind Rhino's fast evolution. We speculate that Kipferl might have evolved out of a necessity for a stabilizer of Rhino that allows it to bind its diverse genomic target loci and to avoid being sequestered by selfish Satellite repeats. While Kipferl's affinity for guanosine-rich sequences optimally opposes the AT-rich Satellite sequences, its low abundance together with the relatively simple DNA motif it recognizes might constitute an optimal level of promiscuous binding across the genome, allowing the targeting of diverse transposon families.

## Materials and methods

### Fly strains and husbandry

All fly stocks were kept at 25 °C with 12 hr dark/light cycles. Fly strains with genotypes, identifiers, and original sources are listed in the Key Resource Table and strains generated for this study are available from VDRC (http://stockcenter.vdrc.at/control/main). For ovary dissections, flies were aged for 2–6 days, and put on apple juice plates with fresh yeast paste for 2 days.

### Generation of transgenic and mutant fly strains

Frame-shift mutant alleles for *CG2678* and *rhino* were generated in isogenised *white* embryos after co-injection of plasmids pBS-Hsp70-Cas9 (Addgene #46294) and pU6-BbsI-chiRNA (Addgene #45946) modified to express sgRNAs. Whole locus *CG2678* deletion for RMCE was achieved following co-injection into ZH-2A(Act5C-Cas9) embryos (derived from Bloomington stock #58492) of plasmids pXZ13 (*Zhang et al., 2014a*) containing 1 kb homology arms around a 3xP3-dsRed marker flanked by attP sites, together with pCFD4 (Addgene 49411) (*Port et al., 2014*) expressing two sgRNAs. sgRNA sequences are given in the Key Resource Table.

Fly strains harboring short hairpin RNA (shRNA) expression cassettes for germline knockdown were generated by cloning shRNA sequences into the Valium-20 vector (*Ni et al., 2011*) modified with a *white* selection marker (oligos: Key Resource Table). Transgenic flies harboring GFP tagged wild-type or engineered Rhino constructs were generated via insertion of desired tag sequences under the control of the *rhino* promoter region and the *vasa* 3'UTR into the attP40 landing site (*Markstein et al., 2008*) in flies harboring a *rhino* frame shift mutation on the same chromosome. Overexpression constructs for *CG2678* were injected as pUASz plasmids (*DeLuca and Spradling, 2018*) containing the full intron-containing sequence of *CG2678* into the attP2 landing site (*Groth et al., 2004*). Transgenic flies expressing TurboID-vhhGFP-3xHA-NLS (cloned from on Addgene #107171) under the *eggless* enhancer were obtained through integration of a mini-white containing plasmid

into the attP40 landing site (*Markstein et al., 2008*). *CG2678* rescue constructs were introduced into the *CG2678* whole locus deletion flies through co-injection of pRVV578 plasmid (Addgene #108279) harboring the endogenous *CG2678* locus flanked by attB sites, with pBS130 (Addgene #26290) for expression of phiC31 integrase. Successful cassette exchange was monitored via loss of dsRed in eyes and the orientation of the inserted construct was verified by PCR. Rescue constructs harbor an internal 3xFLAG tag at residue S161, as neither amino- nor carboxy-terminal tagging of *CG2678* yielded fully functional protein. Exceptions are Kipferl$^{\Delta ZAD}$ and Kipferl$^{GCN4}$ constructs, which harbor an N-terminal 3xFLAG tag. Of note, we found that the CG2678 locus harbors an additional 25 amino acid exon, flanked by 109 and 31 nucleotides of intronic sequence up and downstream, respectively, in several laboratory fly strains. Comparison with protein sequences annotated for other *Drosophila* species confirm the presence of the additional in frame coding sequence, inserted between P347 and K348 of the annotated *melanogaster* CG2678 protein (see also *Figure 2—figure supplement 1*). PCR analysis confirmed the absence of the respective DNA sequence from *iso1* genomic DNA, which served as the basis of the reference genome. The additional sequence is included in the *CG2678* overexpression construct. All RMCE rescue constructs harbor the reference genome locus.

## Generation of endogenous knock-in fly strains

Generation of endogenously tagged lines for *HP1b* and *HP1c* was achieved through co-injections of pU6-BbsI-chiRNA together with pBS donor plasmids containing 1 kb homology arms into embryos from vas-Cas9; attP2 flies or ZH-2A(Act5C-Cas9) embryos for *HP1b* and *HP1c*, respectively. sgRNA sequences are listed in the Key Resource Table.

## Antibody generation

Mouse monoclonal antibodies against His-tagged CG2678 (aa M2-K188) and His-tagged Rhino (full length) were generated by the Max Perutz Labs Antibody Facility. Antigens were cloned in pET-15b, transformed in BL21(DE3) *E. coli*, and purified using Ni-NTA resin (QIAGEN) according to standard protocols. Polyclonal antibodies were raised against a CG2678 peptide (aa R171-I190) at Eurogentec.

## Cell lines

*Drosophila* ovarian somatic cells (OSC) cells were cultured as previously described (*Niki et al., 2006*; *Saito et al., 2009*). Stable OSC lines expressing internally tagged CG2678 under control of the *ubi63E* promoter and an SV40 3'UTR were generated by integration into an RMCE landing site.

## Western blot

Five pairs of ovaries were mechanically disrupted in lysis buffer (20 mM Tris pH 8.0, 1% Triton X-100, 2 mM MgCl$_2$, Benzonase, protease inhibitors) using a plastic pestle. Protein concentration of whole ovary lysate was determined via Bradford assay to allow equal loading, and SDS-PAGE gel electrophoresis was performed according to standard procedures. Primary antibodies were incubated at 4 °C overnight, secondary antibodies for 1 hr at RT and the blots were developed using ECL (BioRad). Antibodies and dilutions are listed in the Key Resource Table.

## RNA fluorescent in situ hybridization

RNA FISH for piRNA clusters *42AB* and *38C*, as well as *HMS-Beagle*, *Max*, *diver*, and *3S18* transposons was performed using Stellaris probes (Biosearch Technologies). Probe sequences are listed in *Supplementary file 3*. RNA FISH for *1.688* and *Rsp* Satellites was performed using a single fluorescent oligo or an in-house labelled probe set of 48 oligos, respectively (*Wei et al., 2021*; *Gaspar et al., 2017*). FISH was performed according to the manufacturers protocol with slight modification. Five ovaries were dissected into ice-cold PBS, fixed at room temperature for 20 min (4% formaldehyde, 0.3% Triton X-100 in PBS), washed three times 5 min at RT (PBS containing 0.3% Triton X-100) and incubated at 4 °C overnight in 70% EtOH to enhance permeabilization. Prior to hybridization, ovaries were rehydrated for 5 min in wash buffer (10% formamide in 2 x SSC). Hybridization was done in 50 µl hybridization buffer (100 mg/ml dextran sulfate and 10% formamide in 2 X SSC) overnight at 37 °C using 0.5 µl Stellaris and *Rsp* FISH probe per sample and a final concentration of 100 nM for the *1.688* FISH oligo. Samples were rinsed two times in wash buffer and then washed in wash buffer two times for 30 min at 37 °C. Ovaries were counterstained for DNA (DAPI 1:5000 in 2 x SSC) for 5 min at RT and

washed two times 5 min with 2 x SSC. Finally, ovaries were mounted on microscopy slides using DAKO mounting medium (Agilent) and equalized at RT for at least 24 hr prior to imaging on a Zeiss LSM 880 inverted Airyscan microscope. Images are given as Z-stack across a maximum of 2 micrometers.

## Immunofluorescence staining of ovaries and testes

Five to 10 ovary pairs or testes were dissected into ice cold PBS and subsequently incubated in fixation solution (4% formaldehyde, 0.3% Triton X-100, 1 x PBS) for 20 min at room temperature with rotation. Fixed ovaries were washed 3 x 5 min in PBX (0.3% Triton X-100, 1 x PBS) and blocked with BBX (0.1% BSA, 0.3% Triton X-100, 1 x PBS) for 30 min, all at room temperature with rotation. Primary antibody incubation was performed by incubation at 4 °C overnight with antibodies diluted in BBX followed by three 5-min washes in PBX. Ovaries were then incubated overnight at 4 °C with fluorophore-coupled secondary antibodies, washed three times in PBX including DAPI in the first wash to stain DNA (1:50,000 dilution). The final wash buffer was carefully removed before addition of ~40 µL DAKO mounting to each sample. The samples were imaged on a Zeiss LSM-880 Axio Imager confocal-microscope and image processing was done using FIJI/ImageJ (*Schindelin et al., 2012*). Images are given as Z-stack across a maximum of 2 micrometer. All relevant antibodies and dilutions are listed in the Key Resource Table.

## Yeast two-hybrid assay

### Screening for Rhino interactors

Yeast two-hybrid screening for Rhino interactors was performed by Hybrigenics Services, S.A.S., Paris, France (https://www.hybrigenics-services.com). The coding sequence for full-length Rhino was cloned into pB27 (derived from pBTM116) (*Vojtek and Hollenberg, 1995*) as an N-terminal fusion to LexA (LexA-Rhi). The construct was sequence verified and used as a bait to screen against a random-primed *Drosophila* ovary cDNA library constructed into pP6 (derived from pGADGH *Bartel et al., 1993*). A mating approach with YHGX13 (Y187 ade2-101::loxP-kanMX-loxP, matα) and L40ΔGal4 (matA) yeast strains was used to screen 65 million interactions as previously described (*Fromont-Racine et al., 1997*). A total of 173 colonies were selected on a medium lacking tryptophan, leucine, and histidine. The prey fragments of the positive clones were amplified by PCR and sequenced at their 5′ and 3′ junctions. The resulting sequences were used to identify the corresponding interacting proteins in the GenBank database (NCBI).

### Validation and interaction mapping

Yeast strains were grown in YPD or SC selective medium at 30 °C. pOAD and pOBD used as backbone for cloning were described in *Miller and Stagljar, 2004* (see Key Resource Table for yeast strains used in this study). Direct protein interactions were probed as described (*Miller and Stagljar, 2004*). In brief, assayed proteins were fused to the activation domain (AD) and DNA-binding domain (DBD) of the Gal4 transcription factor and transformed into yeast strains PJ694A (AD) and PJ694α (DBD). Individually transformed colonies were selected, picked and mated. Interactions were detected upon spotting of a dilution series of mated yeast on selective (-LTH) plates. Parallel plating on non-selective (-LT) plates controlled for presence of both plasmids.

## Biotin proximity labeling

Flies expressing GFP-Rhino or GFP fused to a nuclear localization signal were crossed to flies expressing low levels of TurboID biotin ligase fused to GFP-nanobody and the progeny was kept in cages on apple juice plates and fed with yeast paste containing 100 µM biotin (Sigma) for 16 hr prior to ovary dissection (adapted from *Roux et al., 2018*). A total of 100 µl of ovaries were washed once with ice-cold PBS and dounced (6 times) in 1.2 ml of pre-extraction buffer (10 mM Tris-HCl pH 7.5, 2 mM MgCl$_2$, 3 mM CaCl$_2$, 0.5% NP40, 10% glycerol, cOmplete Protease Inhibitor Cocktail (Roche)) and incubated at 4 °C with nutation for 15 min followed by centrifugation for 5 min at 20,000 g at 4 °C for mild pre-extraction of cytoplasmic contaminants. The nuclear-enriched fraction was resuspended in lysis buffer (50 mM Tris-HCl pH 7.5, 150 mM NaCl, 0.1% SDS, 0.5% Na-Deoxycholate, 1% Triton-X, 1 mM DTT, Benzonase, cOmplete Protease Inhibitor Cocktail (Roche)) and homogenized using first an electric plastic pestle tool (20 s on ice), followed by further douncing (20 times) and incubation at 4 °C with nutation for 2 hr. The lysate was cleared twice by centrifugation at 18,000 g for 10 min

before mixing with 100 µL magnetic Pierce Streptavidin beads (ThermoFischer) pre-equilibrated in lysis buffer, followed by overnight incubation at 4 °C. The beads were washed once with lysis buffer for 10 min at 4 °C, with 2% SDS (10 min at room temperature), and 10 minutes each with wash buffers 1 (50mM HEPES pH 7.5, 500 mM NaCl, 1 mM EDTA, 0.1% Na-Deoxycholate, 1% Triton-X) and 2 (10mM Tris-HCl pH 7.5, 250 mM LiCl, 1 mM EDTA, 0.5% Na-Deoxycholate, 1% NP40) for a minimum of 10 min at 4 °C, followed by five washes without detergent (20 mM Tris-HCl pH 7.5, 137 mM NaCl) before further downstream mass spectrometry analysis.

## Mass spectrometry
Mass spectrometry was carried out as described in *Batki et al., 2019*.

## Scoring of embryo hatching rates
To measure female fertility, 10 virgin females were collected and aged for 2–3 days with at least 24 hr of mating with three *w1118* males. The hatching rate of fertilized eggs laid onto apple juice plates within a period of 4–7 hr was determined 30 hr after egg laying (25 degrees), as percentage of hatched eggs from total. Plates with less than 50 eggs were disregarded in the analysis. Wildtype females were included in every experiment as control.

## Definition and Curation of 1-kb genomic tiles
The four assembled chromosomes of the *Drosophila melanogaster* genome (dm6 assembly) were split into non-overlapping 1-kb tiles. The tiles were annotated by intersection with genomic annotations for piRNA clusters. Tile mappability was determined by intersection with genomic blocks of continuous mappability using bedtools coverage. Tiles with mappability below 25% were excluded from all analyses (2761 1-kb tiles). Further exclusion criteria included a more than threefold deviation from median values for representative input libraries for either of the three wildtype genotypes used in this study (*w1118*, MTD-Gal4 >*w*-sh, *iso1*; affecting 18,268 1-kb tiles), as well as tiles showing strong residual Rhino or CG2678 signal in ChIP-seq libraries prepared from the respective knock out ovaries (20 and 495 tiles, respectively).

## Heterochromatin and euchromatin definitions used in this study
We used ovary H3K9me3 ChIP-seq data to define the extent of pericentromeric heterochromatin and euchromatic chromosome arms. The heavily H3K9me3 covered pericentric regions of the assembled chromosomes, as well as the entire chromosome 4 were classified as heterochromatic, while the rest was annotated as euchromatic. Detailed coordinates can be found in . Small genome contigs not assembled into the four major chromosomes were excluded from all analyses. For 1-kb tile analyses, piRNA clusters *42AB*, *38C*, and *80F* were not included into either category, but were analyzed separately as reference loci.

## ChIP-Seq
ChIP was performed as previously described (*Lee et al., 2006*). In brief, 150 µl of ovaries were dissected into ice-cold PBS, crosslinked with 1.8% formaldehyde in PBS for 10 min at room temperature, quenched with Glycine, rinsed in PBS and flash frozen in liquid nitrogen after removing all PBS. Frozen ovaries were disrupted in PBS using a dounce homogenizer, centrifuged at low speed and the pellet was resuspended in lysis buffer. For ChIP from OSCs 5–10 million cells were crosslinked, quenched, and lysed. Sonication (Bioruptor) resulted in DNA fragment sizes of 200–800 bp. Immunoprecipitation with specific antibodies was done overnight at 4 °C in 350–700 µl total volume using 1/3 to 1/4 of chromatin per ChIP (antibodies are listed in Key Resource Table). Then, 40 µl Dynabeads (equal mixture of Protein G and A, Invitrogen) were added and incubated for 1 hr at 4°. After multiple washes, immuno-precipitated protein-DNA complexes were eluted with 1% SDS, treated with RNAse-A, decrosslinked overnight at 65 °C, and proteins were digested with proteinase K before clean-up using ChIP DNA Clean & Concentrator columns (Zymo Research). Barcoded libraries were prepared according to manufacturer's instructions using the NEBNext Ultra II DNA Library Prep Kit for Illumina (NEB), and sequenced on a HiSeqV4, NextSeq550, or NovaSeqSP (Illumina).

## RNA-Seq

Strand-specific RNA seq was performed as described previously (*Zhang et al., 2012b*). In brief, total RNA was extracted from 5 to 10 ovaries from 7-day-old flies using Trizol (Invitrogen). Total RNA was purified using RNAeasy columns (QIAGEN). Six micrograms of total RNA were subjected to polyA selection and subsequent fragmentation, reverse transcription, and library preparation according to manufacturer's instructions using the NEBNext Ultra DNA Library Prep Kit for Illumina (NEB) for sequencing on an Illumina NovaSeqSP instrument.

## Small RNA-Seq

Small RNA cloning was performed as described in *Grentzinger et al., 2020*. In brief, ovaries were lysed and Argonaute- sRNA complexes were isolated using TraPR ion exchange spin columns. sRNAs were subsequently purified using Trizol and subjected to ligations of 3′ and 5′ barcoded adapters containing 4 random nucleotides at the ends to reduce ligation biases, reverse transcribed, PCR amplified, and sequenced on an Illumina NextSeq550 instrument.

## Computational analysis

### ChIP-Seq analysis

ChIP-seq reads were trimmed to remove the adaptor sequences and to adjust all reads to 50 bp irrespective of sequencing mode. Reads were mapped to the dm6 genome using Bowtie (version.1.3.0, settings: -f -v 3 a `--best --strata --sam`), allowing up to three mismatches. Genome unique reads were mapped to 1-kb tiles and a pseudocount of 1 was added after normalization to library depth, before enrichment over input values were determined. Each ChIP-seq sample was adjusted with a correction factor determined from median input levels and median background levels to reach median background enrichment of 1 to correct for unequal ChIP efficiency. To classify genomic regions into Rhino domains and non-Rhino domains, we used a binary cutoff of 4-fold enrichment calculated from two independent replicate experiments of the relevant wildtype genotypes. This cutoff corresponds to a p-value of <0.05. Kipferl-only 1-kb tiles were those that had no Rhino enrichment (below 4-fold) and that were significantly enriched in Kipferl over Rhino (Z-score=3). Replicates were averaged for genomic 1-kb tile analyses.

### ChIP-seq peak calling

We used MACS2 (*Zhang et al., 2008*) with `--broad --broad-cutoff` 0.1 for Kipferl and Rhino in wildtype ovaries due to the broad extent of Rhino/Kipferl domains. The 'narrow peak' setting was used for the remaining experiments. Peaks mapping to genomic contigs outside the four main chromosomes were discarded, and peaks were filtered for a score of 50 (broad peaks; $p < 10^{-5}$) and 30 (narrow peaks; $p < 10^{-3}$). Kipferl-Rhino shared versus Kipferl-only peaks were distinguished by intersection of narrow peaks called for Kipferl in two independent replicate experiments of *rhino* mutant ovaries, with broad peaks called for Rhino in two independent replicate experiments of $w^{1118}$ ovaries using bedtools intersect with -u -f 0.75 for shared domains and -v for Kipferl-only domains. Rhino-independent Kipferl peaks that were detected independently in two replicate experiments were grouped into heterochromatic and euchromatic by intersection with heterochromatin coordinates outlined above. Kipferl DNA binding motifs were recovered from the top ~3000 summits of Rhino-independent Kipferl peaks (achieved through a score cutoff of 7 on summits) using HOMER (*Heinz et al., 2010*). Heatmaps display one representative replicate and were produced through deeptools (*Ramírez et al., 2016*).

### ChIP-seq analysis on transposon consensus sequences

Genome mapping reads longer than 23 nucleotides were mapped to TE consensus sequences (*Supplementary file 4*) using bowtie (v.1.3.0; settings: -f -v 3 a `--best --strata --sam`) allowing up to three mismatches. Reads mapping to multiple elements were assigned to the best mapping position. Reads mapping to multiple positions were randomly distributed. Library depth normalized ChIP and input reads, respectively, were averaged over all nucleotide positions of each element to give one value per element. ChIP-seq enrichment was calculated after adding a pseudo count of 1 and adjusted using sample-specific correction factors determined from background 1-kb tiles to reach median background enrichments of 1. Corrected per-base enrichment was calculated for TE ChIP-seq profiles and replicates were averaged.

## Motif instances

Occurrences of Kipferl DNA binding motifs were determined using PWMScan (ccg.epfl.ch/pwmtools/pwmscan.php) on the dm6 genome and on TE consensus sequences. For display in heatmaps, cumulative motif counts on both genomic strands were intersected with non-overlapping 100-bp windows.

## smallRNA-Seq Analysis:

Raw reads were trimmed for linker sequences and the 4 random nucleotides flanking the small RNA before mapping to the *Drosophila melanogaster* genome (dm6), using Bowtie (version.1.3.0, settings: `-f -v 3 a --best --strata --sam`) with 0 mismatch allowed. Genome mapping reads were intersected with Flybase genome annotations (r6.40) using Bedtools to allow the removal of reads mapping to rRNA, tRNA, snRNA, snoRNA loci and the mitochondrial genome. For TE mappings, all genome mappers were used allowing no mismatches. Reads mapping to multiple elements (*Supplementary file 4*) were assigned to the best match. Reads mapping equally well to multiple positions were randomly distributed. Libraries were normalized to 1 Mio miRNA reads. For the 1-kb tiles analysis, a pseudocount of 1 was added after normalization to library depth and correction for the mappability of the respective 1-kb tile. Tiles with fewer than 10 mapping piRNAs in all libraries were disregarded for sRNA analysis to avoid distortion due to very low abundant piRNAs. For calculation of piRNAs mapping to TEs, sense and antisense piRNAs were kept separate, and counts were normalized to TE length. For classification of tiles and transposons into somatic, Rhino-independent, and Rhino-dependent source loci, the soma index was determined as the log2 ratio of somatic (Piwi-IP in *piwi* GLKD, normalized to library depth) and germline (GL-Piwi IP, normalized to library depth) piRNAs mapping to each tile or TE (*Mohn et al., 2014*). Classification by Kipferl-dependency of TEs was achieved by a binary cutoff of at a twofold reduction in antisense piRNA levels in *kipferl* knock down compared to control.

## RNA-Seq analysis

For the RNAseq analysis, genome matching reads (STAR v2.7.10a; settings: `--outSAMmode NoQS --readFilesCommand cat --alignEndsType Local --twopassMode Basic --outReadsUnmapped Fastx --outMultimapperOrder Random --outSAMtype SAM --outFilterMultimapNmax 1000 --winAnchorMultimapNmax 2000 --outFilterMismatchNmax 3 --seedSearchStartLmax 30 --alignSoftClipAtReferenceEnds No --outFilterType BySJout --alignSJoverhangMin 15 --alignSJDBoverhangMin 1`) were randomized in order and quantified using Salmon (v.1.7.0; settings: --dumpEqWeights --seqBias --gcBias --useVBOpt --numBootstraps 100 | SF --incompatPrior 0.0 --validateMappings). For the analysis we used the FlyBase transcriptome (r6.40) which has been masked for sequences similar to transposons. To include both strands of transposons in the analysis, TE-consensus sequences were added to the FlyBase transcriptome in sense and antisense orientation. For gene expression visualization Salmon results were further processed to GeTMM values using edgeR (v3.34.0). For differential gene expression analysis Salmon results were processed using DeSeq2 (v1.32.0).

## TE insertion Calling

Euchromatic TE insertions were extracted from insertions called previously (*Mohn et al., 2014*), and nearby insertions were merged using bedtools merge -d 100, as these mostly corresponded to the same insertions called at slightly different positions. Further, all insertions overlapping UCSC repeat masker track annotations were discarded. For analysis of non-Kipferl nucleation site containing, but Rhino-dependent TEs, the resulting list was subsequently filtered for elements with no Kipferl enrichment in *rhino* knock out ovaries and at least 10-fold difference in piRNA levels between control and MTD-Gal4 mediated *rhino* germline knock down. This retrieved 285 euchromatic solo TE insertion sites in the genome of our experimental MTD-Gal4 strains (*Supplementary file 5*).

## Acknowledgements

We thank the VBCF core facilities (NGS, VDRC, Protein Chemistry) as well as the IMBA/GMI/IMP BioOptics facility for excellent support, and the IMBA Fly House for generating transgenic and CRISPR-edited

fly lines. Mia Levine gave invaluable input on the analysis of positive selection at the *kipferl* locus. The Max Perutz Laboratories Monoclonal Antibody facility generated the CG2678 and Rhino hybridoma cell lines. Steven DeLuca shared the pUASz plasmid. Sarah Barnes helped with cloning of the Kipfer-l$^{\Delta ZnF\#4}$ construct, Bernardo Almeida gave advice on motif analyses, and Maria Novatchkova supported bioinformatic analyses. We thank the Brennecke laboratory for help throughout this project and A Larracuente, F Mohn, J Schnabl, and S Barnes for comments on the manuscript. This research was funded by the Austrian Academy of Sciences, the European Community (ERC-2015-CoG-682181) and the Austrian Science Fund (F4303 and W1207). LB was funded by a Boehringer Ingelheim Fond PhD fellowship. CY was funded by a VIP2 postdoc fellowship as part of the EU Horizon 2020 research and innovation program (Marie Skłodowska-Curie grant no. 847548).

## Additional information

### Funding

| Funder | Grant reference number | Author |
|---|---|---|
| European Research Council | ERC-2015-CoG-682181 | Julius Brennecke |
| Marie Curie | 847548 | Changwei Yu |
| Austrian Science Fund | F4303 and W1207 | Julius Brennecke |
| Boehringer Ingelheim Fonds | | Lisa Baumgartner |

The funders had no role in study design, data collection and interpretation, or the decision to submit the work for publication.

### Author contributions

Lisa Baumgartner, Conceptualization, Data curation, Formal analysis, Funding acquisition, Validation, Investigation, Visualization, Methodology, Writing – original draft, Project administration, Writing – review and editing; Dominik Handler, Data curation, Software, Formal analysis; Sebastian Wolfgang Platzer, Investigation, Methodology; Changwei Yu, Resources, CY established and optimized the versatile GFP-nanobody based biotin proximity labeling assay in flies; Peter Duchek, Methodology, PD designed and generated all CRISPR engineered flies; Julius Brennecke, Conceptualization, Resources, Supervision, Funding acquisition, Writing – original draft, Project administration, Writing – review and editing

### Author ORCIDs

Lisa Baumgartner http://orcid.org/0000-0001-7769-1274
Dominik Handler http://orcid.org/0000-0002-1059-4960
Sebastian Wolfgang Platzer http://orcid.org/0000-0002-2518-6273
Changwei Yu http://orcid.org/0000-0002-0119-8616
Julius Brennecke http://orcid.org/0000-0002-5141-0814

### Ethics

All work described in this paper uses Drosophila melanogaster as model organism. Relevant safety procedures in terms of handling transgenic flies are in place.

### Decision letter and Author response

Decision letter https://doi.org/10.7554/eLife.80067.sa1
Author response https://doi.org/10.7554/eLife.80067.sa2

## Additional files

### Supplementary files

• Supplementary file 1. Chromosomal coordinates of major heterochromatic and euchromatic compartments used in this study.

• Supplementary file 2. Putative protein-protein interactors of Rhino identified through a yeast two-hybrid screen.

• Supplementary file 3. Sequences of RNA FISH probes used in this study.

• Supplementary file 4. Consensus sequences of transposable elements analyzed in this study.

• Supplementary file 5. Chromosomal coordinates of stand-alone transposon insertions identified in the MTD-Gal4 background that are absent in the reference genome.

• MDAR checklist

### Data availability

Sequencing data sets have been deposited to the NCBI GEO archive (GSE202468). All fly strains generated for this study are available via the VDRC. All genome-wide sequencing data (ChIP-seq, small RNA-seq, RNA-seq) can be browsed at https://genome-euro.ucsc.edu/s/balisa/dm6_elife220506.

The following dataset was generated:

| Author(s) | Year | Dataset title | Dataset URL | Database and Identifier |
|---|---|---|---|---|
| Brennecke J | 2022 | The *Drosophila* ZAD zinc finger protein Kipferl guides Rhino to piRNA clusters | http://www.ncbi.nlm.nih.gov/geo/query/acc.cgi?acc=GSE202468 | NCBI Gene Expression Omnibus, GSE202468 |

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

# Appendix 1

## Appendix 1—key resources table

| Reagent type (species) or resource | Designation | Source or reference | Identifiers | Additional information |
|---|---|---|---|---|
| Antibody | anti-CG2678#2 (Rabbit polyclonal) | This paper | CG2678#2_4P39glyc, raised against Kipferl peptide R171-I190 | Anti-Kipferl polyclonal antibody, available from Brennecke lab; ChIP (7 µL per IP) |
| Antibody | anti-CG2678 M4 (Mouse monoclonal) | This paper | M4 3A6-3E1-E11, raised against Kipferl amino acids M2-K188 | Anti-Kipferl monoclonal WB antibody, available from Brennecke lab; WB (1:250) |
| Antibody | anti-CG2678 M3 (Mouse monoclonal) | This paper | M3 2 C5-3C3, raised against Kipferl amino acids M2-K188 | Anti-Kipferl monoclonal IF antibody, available from Brennecke lab; IF (1:500) |
| Antibody | anti-Rhino (Mouse monoclonal) | This paper | 6B7-F2, raised against full length denatured Rhino protein | Anti-Rhino monoclonal WB antibody, available from Brennecke lab; WB (1:10) |
| Antibody | anti-Rhino (Rabbit polyclonal) | *Mohn et al., 2014* | Rhino#1_3573gly | ChIP (5 µL per IP), IF (1:1000) |
| Antibody | anti-Deadlock (Mouse monoclonal) | *Andersen et al., 2017* | 5B5-6D7-3H10 | IF (1:100) |
| Antibody | anti-Nxf3 (Mouse monoclonal) | *ElMaghraby et al., 2019* | 8E4-F1 | IF (1:50) |
| Antibody | Histone H3K9me3 antibody (Rabbit polyclonal) | Active motif | ID_source:39161 | ChIP (5 µL per IP), IF (1:100) |
| Antibody | Anti-Histone H3 (di methyl K9) (mouse monoclonal) | Abcam | ID_source:mAbcam1220 | ChIP (5 µL per ChIP) |
| Antibody | C1A9 HP1 (Su(var)2–5) (Mouse monoclonal) | DSHB | ID_source:C1A9 | ChIP (5 µL per IP), IF (1:100) |
| Antibody | ANTI-FLAG(R) M2 (Mouse monoclonal) | Sigma Aldrich | ID_source:F1804-1MG | ChIP (2 µL per IP), IF (1:2000) |
| genetic reagent (*D. melanogaster*) | w<sup>1118</sup>;;; | Bloomington stock 3605 | w<sup>1118</sup> | wildtype, cultivated in our lab for several years |
| genetic reagent (*D. melanogaster*) | w;;; | Susan Celniker | iso1 | wildtype |
| genetic reagent (*D. melanogaster*) | Cog-GAL4; NGT-GAL4; nos-GAL4; | Bloomington stock 31777 | Referred to as 'maternal triple driver' (MTD) | Gal4 driver |
| genetic reagent (*D. melanogaster*) | w;; pW20>w_sh[attP2]/TM3, Sb; | *Mohn et al., 2014*, VDRC-ID 313772 | white (CG2759) | RNAi line |
| genetic reagent (*D. melanogaster*) | w;; pW20>rhi_sh[attP2]/TM3, Sb; | *Mohn et al., 2014*, VDRC-ID 313156 | Rhino (CG10683) | RNAi line |
| genetic reagent (*D. melanogaster*) | w;; pW20>CG2678_sh2 [attP2]/TM3, Sb; | this paper | Kipferl (CG2678) | Kipferl RNAi line, available from VDRC; sh-oligo sequence: ctagcagtCTCGAAGG CTTTCATGCGTAA tagttatattcaagcata TTACGCATGAAA GCCTTCGAGgcg |
| genetic reagent (*D. melanogaster*) | w;; pGLKD >piwi_sh2 [attP2]/TM3,Sb; | *Senti and Brennecke, 2010*, VDRC-ID 313199 | Piwi (CG6122) | RNAi line |
| genetic reagent (*D. melanogaster*) | w;; CG2678[Δ1] (dsRed+)/TM3,Sb; | this paper | Kipferl (CG2678) | Kipferl mutant allele, available from VDRC; LB1-RMCEm31 |
| genetic reagent (*D. melanogaster*) | w;; CG2678[Δ2](dsRed+)/TM3,Sb; | this paper | Kipferl (CG2678) | Kipferl mutant allele, available from VDRC; LB1-RMCEm21; has aberrant deregulation of blood transposon |

*Appendix 1 Continued on next page*

*Appendix 1 Continued*

| Reagent type (species) or resource | Designation | Source or reference | Identifiers | Additional information |
|---|---|---|---|---|
| genetic reagent (*D. melanogaster*) | *w;; CG2678[fs1]/TM3,Sb;* | this paper | Kipferl (CG2678) | Kipferl mutant allele, available from VDRC; LB1-FSm52; indel (–7); sequence CCTGCGT CCTGGCCGTGC------- TTTCCGGTTCAAGT GGCAAAGCGAGCAGAG |
| genetic reagent (*D. melanogaster*) | *w;; CG2678RMCE[S161-3xFLAG]/TM3,Sb;* | this paper | Kipferl (CG2678) | Kipferl tagged construct (RMCE), available from VDRC; wildtype 3xFLAG tagged rescue construct inserted into LB1-RMCEm31 |
| genetic reagent (*D. melanogaster*) | *w; prhino >3xFLAG/V5/ Precission/GFP-Rhino[attP40],rhi[g2m11]/CyO; CG2678[Δ1]*(dsRed+), *nxf3[A2-2]/TM3,Sb;* | this paper | Rhino, Kipferl, Nxf3 | mutant allele combination, available from VDRC; Nxf3[A2-2] allele published in ***ElMaghraby et al., 2019*** |
| genetic reagent (*D. melanogaster*) | *w; prhino >3xFLAG/V5/Precission/GFP-Rhino[attP40],rhi[g2m11]/CyO;;* | this paper | Rhino (CG10683) | tagged Rhino construct, available from VDRC |
| genetic reagent (*D. melanogaster*) | *w; prhino >3xFLAG/V5/Precission/ GFP-Rhino[attP40],rhi[g2m11]/CyO; CG2678[Δ1]*(dsRed+)/TM3,Sb; | this paper | Rhino, Kipferl | mutant allele combination, available from VDRC |
| genetic reagent (*D. melanogaster*) | *w; rhi[18-7]/CyO;;* | ***Andersen et al., 2017***, VDRC-ID 313488 | Rhino (CG10683) | mutant allele |
| genetic reagent (*D. melanogaster*) | *w; rhi[g2m11]/CyO;;* | this paper | Rhino (CG10683) | Rhino mutant allele, available from VDRC; indel –7; seq: ATGTCT CGCAACCA-------cc-A ATCTTGGTCTGGTC GATGCACCGCCTAATG |
| genetic reagent (*D. melanogaster*) | *w;; pUASz >CG2678[attP2];* | this paper | Kipferl (CG2678) | tagged Kipferl construct, available from VDRC; intron containing CG2678 sequence including non-mapped exon3 |
| genetic reagent (*D. melanogaster*) | *w;; CG2678RMCE [ΔZnFarray1-S161-3xFLAG]/TM3,Sb;* | this paper | Kipferl (CG2678) | tagged Kipferl construct (RMCE), available from VDRC; 3xFLAG tagged ZnF array 1 deletion construct inserted into LB1-RMCEm31 |
| genetic reagent (*D. melanogaster*) | *w;; CG2678RMCE [ΔZnFarray2-S161-3xFLAG]/TM3,Sb;* | this paper | Kipferl (CG2678) | tagged Kipferl construct (RMCE), available from VDRC; 3xFLAG tagged ZnF array 2 deletion construct inserted into LB1-RMCEm31 |
| genetic reagent (*D. melanogaster*) | *w;; CG2678RMCE [3xFLAG-ΔZAD]/TM3,Sb;* | this paper | Kipferl (CG2678) | tagged Kipferl construct (RMCE), available from VDRC; 3xFLAG tagged ZAD deletion construct inserted into LB1-RMCEm31 |
| genetic reagent (*D. melanogaster*) | *w;; CG2678RMCE[3xFLAG-ZAD::GCN4]/ TM3,Sb;* | this paper | Kipferl (CG2678) | tagged Kipferl construct (RMCE), available from VDRC; 3xFLAG tagged ZAD GCN4 replacement construct inserted into LB1- RMCEm31 |
| genetic reagent (*D. melanogaster*) | *w;; CG2678RMCE [3xFLAG-ouibZAD]/TM3,Sb;* | this paper | Kipferl (CG2678) | Tagged Kipferl construct (RMCE), available from VDRC; 3xFLAG tagged ZAD replacement construct inserted into LB1-RMCEm31 |
| genetic reagent (*D. melanogaster*) | *w;; CG2678RMCE[ΔZnF#4-S161-3xFLAG]/ TM3,Sb;* | this paper | Kipferl (CG2678) | tagged Kipferl construct (RMCE), available from VDRC; 3xFLAG tagged ZnF4 deletion construct inserted into LB1-RMCEm31 |

*Appendix 1 Continued on next page*

*Appendix 1 Continued*

| Reagent type (species) or resource | Designation | Source or reference | Identifiers | Additional information |
|---|---|---|---|---|
| genetic reagent (*D. melanogaster*) | *w*; p*rhino* >3xFLAG/V5/Precission/GFP-Rhino(CSD::GCN4)[attP40],*rhi[g3m13]*/CyO;; | this paper | Rhino (CG10683) | tagged Rhino construct, available from VDRC; Rhino indel –14; seq: TGGGCGTCCCCAGG------------- ---AGCGGTTTTCCGAA CGAGAACAACACC, Rhino CSD was replaced by the Gcn4 dimerization domain, homozygous not viable, crossed to w; rhi[18-7]/CyO;; for experiments |
| genetic reagent (*D. melanogaster*) | *w*; p*rhino* >3xFLAG/V5/Precission/GFP-Rhino(art.hinge)[attP40],*rhi[g3m13]*/CyO;; | this paper | Rhino (CG10683) | tagged Rhino construct, available from VDRC; Rhino indel –14; seq: TGGGCGTCCCCAGG-------- -------- AGCGGTTTTCCGA ACGAGAACAACACC, Rhino hinge was replaced by a scrambled amino acid sequence of the same length, homozygous not viable, crossed to w; rhi[18-7]/CyO;; for experiments |
| genetic reagent (*D. melanogaster*) | *w*;; p*rhino* >3xFLAG/V5/Precission/GFP-Su(var)2-5[attP2]; | this paper | HP1a (CG8409) | tagged HP1a construct, available from VDRC |
| genetic reagent (*D. melanogaster*) | *w*, 3xFLAG/V5/Precission/GFP-HP1b;;; | this paper | HP1b (CG7041) | Endogenously tagged HP1b, available from VDRC |
| genetic reagent (*D. melanogaster*) | *w*;; 3xFLAG/V5/Precission/GFP-HP1c; | this paper | HP1c (CG6990) | Endogenously tagged HP1c, available from VDRC |
| genetic reagent (*D. melanogaster*) | *w*; p*eggless* >TurboID-linker-vhhGFP-3xHA-NLS[attP40]/CyO;; | this paper | - | Transgenic construct, available from VDRC; TurboID biotin ligase fused to GFP nanobody |
| genetic reagent (*D. melanogaster*) | *w*;; 3xFLAG/V5/Precission/GFP(replacing CG13741 CDS)-NLS[attP2]/TM3,Sb; | **ElMaghraby et al., 2019** | Boot (CG13741) | tagged construct; nuclear GFP used as contol |
| Sequence-based reagent | Rhino_g2 | This paper | CRISRP guide RNA (indel) | GACCAAGATTTGGTCGCTGA |
| Sequence-based reagent | Rhino_g3 | This paper | CRISRP guide RNA (indel) | GTCCCCAGGTTCTGGTGAAG |
| Sequence-based reagent | CG2678_g1 | This paper | CRISRP guide RNA (RMCE left) | GTACAAATGATCAGTGCGA |
| Sequence-based reagent | CG2679_g2 | This paper | CRISRP guide RNA (RMCE right) | GAAGGCATTAAGTAGCATG |
| Sequence-based reagent | CG2678_g3 | This paper | CRISRP guide RNA (indel) | GAACCGGAAAGCATTCTGCA |
| Sequence-based reagent | HP1b_g2 | This paper | CRISRP guide RNA (N-terminal endogenous tagging) | CACAATGGCCGAATTCTCAG |
| Sequence-based reagent | HP1c_g2 | This paper | CRISRP guide RNA (N-terminal endogenous tagging) | GATGCGCTCCACCACGAAGT |
| Strain, strain background (*Saccharomyces cerevisiae*) | *Matα, trp1-901, leu2-3, 112, ura3-52, his3-200, gal4Δ, gal80Δ, GAL2-ADE2, LYS2::GAL1-HIS3, met2::GAL7-lacZ* | **Mohn et al., 2014** | | |
| Strain, strain background (*Saccharomyces cerevisiae*) | *MatA, trp1-901, leu2-3, 112, ura3-52, his3-200, gal4Δ, gal80Δ, GAL2-ADE2, LYS2::GAL1-HIS3, met2::GAL7-lacZ* | **Mohn et al., 2014** | | |

*Appendix 1 Continued on next page*

*Appendix 1 Continued*

| Reagent type (species) or resource | Designation | Source or reference | Identifiers | Additional information |
|---|---|---|---|---|
| Strain, strain background (*Saccharomyces cerevisiae*) | pOAD | **Mohn et al., 2014** | | in YGS2 (MatA) |
| Strain, strain background (*Saccharomyces cerevisiae*) | pOAD-Rhino | **Mohn et al., 2014** | | in YGS2 (MatA) |
| Strain, strain background (*Saccharomyces cerevisiae*) | pOBD-Rhino | **Mohn et al., 2014** | | in YGS1 (Matα) |
| Strain, strain background (*Saccharomyces cerevisiae*) | pOAD-HP1a | **Mohn et al., 2014** | | in YGS2 (MatA) |
| Strain, strain background (*Saccharomyces cerevisiae*) | pOAD-Rhino(1-100) | **Mohn et al., 2014** | | in YGS2 (MatA) |
| Strain, strain background (*Saccharomyces cerevisiae*) | pOAD-Rhino(101-300) | **Mohn et al., 2014** | | in YGS2 (MatA) |
| Strain, strain background (*Saccharomyces cerevisiae*) | pOAD-Rhino(301-418) | **Mohn et al., 2014** | | in YGS2 (MatA) |
| Strain, strain background (*Saccharomyces cerevisiae*) | pOBD-CG2678-ZF4nolinker | this paper | | Y2H construct, available from Brennecke lab; in YGS1 (Matα) |
| Strain, strain background (*Saccharomyces cerevisiae*) | pOBD-CG2678-190-311 | this paper | | Y2H construct, available from Brennecke lab; in YGS1 (Matα) |
| Strain, strain background (*Saccharomyces cerevisiae*) | pOAD-CG2678FL | this paper | | Y2H construct, available from Brennecke lab; in YGS2 (MatA) |
| Strain, strain background (*Saccharomyces cerevisiae*) | pOAD-CG2678frag1 | this paper | | Y2H construct, available from Brennecke lab; in YGS2 (MatA) |
| Strain, strain background (*Saccharomyces cerevisiae*) | pOAD-CG2678frag2 | this paper | | Y2H construct, available from Brennecke lab; in YGS2 (MatA) |
| Strain, strain background (*Saccharomyces cerevisiae*) | pOAD-CG2678frag3 | this paper | | Y2H construct, available from Brennecke lab; in YGS2 (MatA) |
| Strain, strain background (*Saccharomyces cerevisiae*) | pOAD-CG2678frag4 | this paper | | Y2H construct, available from Brennecke lab; in YGS2 (MatA) |
| Strain, strain background (*Saccharomyces cerevisiae*) | pOAD-CG2678frag5 | this paper | | Y2H construct, available from Brennecke lab; in YGS2 (MatA) |
| Strain, strain background (*Saccharomyces cerevisiae*) | pOAD-CG2678-ZF1-2 | this paper | | Y2H construct, available from Brennecke lab; in YGS2 (MatA) |

*Appendix 1 Continued on next page*

*Appendix 1 Continued*

| Reagent type (species) or resource | Designation | Source or reference | Identifiers | Additional information |
|---|---|---|---|---|
| Strain, strain background (*Saccharomyces cerevisiae*) | pOAD-CG2678-ZF1-3 | this paper | | Y2H construct, available from Brennecke lab; in YGS2 (MatA) |
| Strain, strain background (*Saccharomyces cerevisiae*) | pOAD-CG2678ZF2-4 | this paper | | Y2H construct, available from Brennecke lab; in YGS2 (MatA) |
| Strain, strain background (*Saccharomyces cerevisiae*) | pOAD-CG2678ZF3-4 | this paper | | Y2H construct, available from Brennecke lab; in YGS2 (MatA) |
| Strain, strain background (*Saccharomyces cerevisiae*) | pOAD-CG2678ZF4nolinker | this paper | | Y2H construct, available from Brennecke lab; in YGS2 (MatA) |
| Strain, strain background (*Saccharomyces cerevisiae*) | pOAD-OuibZAD | this paper | | Y2H construct, available from Brennecke lab; in YGS2 (MatA) |
| Strain, strain background (*Saccharomyces cerevisiae*) | pOAD-Rhi19-85 | this paper | | Y2H construct, available from Brennecke lab; in YGS2 (MatA) |
| Strain, strain background (*Saccharomyces cerevisiae*) | pOAD-CG2678-190-311 | this paper | | Y2H construct, available from Brennecke lab; in YGS2 (MatA) |
| Strain, strain background (*Saccharomyces cerevisiae*) | pOBD-CG2678frag1 | this paper | | Y2H construct, available from Brennecke lab; in YGS1 (Matα) |
| Strain, strain background (*Saccharomyces cerevisiae*) | pOAD-HP1CD | this paper | | Y2H construct, available from Brennecke lab; in YGS2 (MatA) |

