## [Editor Report]

This paper reports fundamental insights into host-transposon interactions, and more specifically how specific genomic loci may be elected for producing anti-transposon piRNAs in *Drosophila*. The authors provide here compelling evidence that Kipferl, a ZAD zinc-finger protein, helps guide Rhino to G-rich motifs present in piRNA-producing loci in the female germline, demonstrating for the first time the existence of sequence-specific factors in piRNA biogenesis. The findings are of broad interest to the fields of heterochromatin and transposon biology.

---

## [Decision Letter]

**Decision letter after peer review:**

Thank you for submitting your article "The *Drosophila* ZAD zinc finger protein Kipferl guides Rhino to piRNA clusters" for consideration by *eLife*. Your article has been reviewed by 3 peer reviewers, and the evaluation has been overseen by a Reviewing Editor and Kevin Struhl as the Senior Editor. The following individual involved in review of your submission has agreed to reveal their identity: Zhao Zhang (Reviewer #2).

Essential revisions:

The reviewers have greatly enjoyed reading the manuscript and recognized the quality, novelty, and impact of the findings, supported by a large breadth of complementary approaches. How active piRNA-producing loci can emerge from heterochromatin and how they can be distinguished from other heterochromatic loci is an essential question, which more generally touches upon the question of the transcriptional activity of supposedly inert heterochromatin. Importantly, the identification of Kipferl may lead the path for the identification of other sequence-specific determinants of piRNA-producing loci, as H3K9me3-enrichment seems unlikely to be the sole feature allowing piRNA loci to be transcribed.

We identified five points that need further attention and are listed below. Most of them involve toning down some of the claims and /or integrating prior knowledge in the novel model, such as the epistatic position of Kipferl towards maternally deposited piRNAs in the specification of piRNA-producing loci. Please address them in a revised version of your manuscript. In addition, answer the individual comments of the reviewers in a detailed rebuttal letter.

1 – Kipferl was identified as a Rhino partner by yeast two-hybrid. However, independent confirmation by co-IP experiments seems to be missing. Please provide such experimental validation.

2 – Regarding the potential position of Kipferl at the interface of a genetic conflict, reviewers were interested to know whether Kipferl was conserved outside of *melanogaster* and more particularly, in *simulans*, and if so, whether some rapidly evolving amino-acids could be identified between the two species.

3 – The claim that Kipferl directly binds to DNA should be toned down, as recruitment to Rhino-occupied loci through protein/protein interactions is equally possible at that stage, unless biochemical evidence is provided.

4 – Please integrate the role of maternally deposited piRNAs into the model of Rhino recruitment to piRNA loci, and where they stand regarding Kipferl. For example, are these two modes of recruitment temporally separated (maternal piRNAs being more in stem/progenitor cells of the female germline, while Kipferl may be acting in more differentiated female germ cells, and be therefore involved in maintenance rather than original specification)? Do maternal piRNAs and Kipferl activate different piRNA clusters? Or do they act redundantly? No specific experiments are required here, but the view(s) of the authors on that matter would be very useful to share, as maternal piRNA-dependent specification of piRNA-producing loci is a well-accepted model.

5 – The authors propose that Kipferl acts upstream from Rhino but the hierarchy between Kipferl and Rhino seems more complicated than this simple model, as Kipferl binding at piRNA loci can also be reduced or even completely gone in Rhino mutants. The authors indeed discuss that Rhino is required for ectopic Kipferl spreading to Satellite loci, but it seems that Rhino is also required to some extent for Kipferl to bind at canonical piRNA loci. The authors are not required to carry out a tethering experiment-as suggested by reviewer #2- but the text should be modulated accordingly.

Figure revisions:

– Please provide statistics in violin and box plots. Currently, the reader may have the impression that changes may be present but non-significant.

– Genomic coordinates have to be provided on UCSC genome tracks.

– Please homogenize your scales in figures of genome tracks, particularly when comparing different genotypes for a given locus (see Figures 1C and 5C for example).

*Reviewer #1 (Recommendations for the authors):*

1. Without Kipferl, H3K9me3 seems to mildly (but visibly) increase (Figure 3G) while H3K9me2 seems to mildly (but visibly) decrease (Figure S3G). Is there a statistically significant difference between the two genotypes for K9me2/3? Is there a correlation between change in H3K9me2/3 and change in Rhino binding and what is their correlation coefficient?

2. In Abstract, Intro and Discussion, authors describe the exciting possibility that Rhino enrichment on satDNA in *kipferl-/-* could reflect a conflict. Given that Rhino binds satDNA in wildtype, what about the alternative possibility that it is merely a passive consequence of Rhino having to re-localize to Kipferl-independent loci in *kipferl-/-*? Or is there a disproportionate, higher-than-expected enrichment onto satDNA, with respect to *42AB* and *38C*, upon loss of Kipferl? Is there an increase in *42AB* and *38C* transcripts in *kipferl-/-* by RNA FISH? What makes up the remaining 79% piRNAs when 21% are satDNA piRNAs at line 307?

3. It is unclear which sequencing experiment is done with replicates and how many replicates, which should be clearly described. For replicated experiment, is a representative dataset OR the average of replicates shown? In sequencing experiments without replicates, how could one be convinced of the result?

4. In all confocal images, is only one z-slice shown? Other than Figure 3C, Rhino seems to have additional foci besides the pastry domain at the absence of Kipferl (e.g., Figures 6D, S3C, S3H). Showing the z-projection for single nurse cells might make statements about Rhino localization more convincing, especially the claim that *kipferl, nxf3* double mutants restore Rhino foci that are absent in *kipferl-/-.*

5. The correlation coefficient with respective p-value should be reported whenever a correlation argument is made throughout the manuscript. e.g., lines 161, 347 and throughout Figures7 and S7.

6. Text description should not exaggerate the result. e.g., "every" at line 556. Text often says Kipferl is important for most/majority of Rhino domains, could it be supported by actual numbers (# out of total #, either # domain or # 1kb tiles)? Given that most active piRNA loci *42AB* and *38C*, as well as telomeric transposons, are all Kipferl-independent, one wonders if Kipferl really controls most/majority of Rhino domains. Since there are many Rhino-only and Kipferl-only domains/1kb tiles, listing actual numbers helps readers contextualize the relationship between Rhino and Kipferl as well as the prevalence of exceptions.

*Reviewer #2 (Recommendations for the authors):*

This is a compelling story to establish the function of Kipferl in recruiting Rhino to the piRNA-producing loci in *Drosophila* ovarian genome. I enjoyed reading and considering it in the past few days. I felt that there are some questions still unanswered--particularly how Rhino is recruited to *42AB*. However, given the authors already generated massive amount of data and the complexity of Rhino recruiting mechanisms, I don't think it would be fair to ask for an answer here. Here are a few suggestions:

1. I think the authors should comment on the potential homolog of Kipferl in other *Drosophila* species. Rhino appears to be rapidly evolving, even simulans Rhino could not be compatible with the *melanogaster* piRNA machineries. Does Kipferl homolog exist in simulans. If yes, how different it is with the *melanogaster* Kipferl, especially the amino acids involving in Rhino and DNA binding. Answering these questions could potentially broaden the impact of this story.

2. Line159, H3K9me3 or H3K9me2?

3. I am confused about the message from Line 365-367. Do you mean Rhino peaks towards the periphery?

4. Line 452: motives or motifs.

5. The last paragraph is way too long. It is good to only deliver one message per paragraph.

*Reviewer #3 (Recommendations for the authors):*

The manuscript, although well written, is very dense with a lot of figures which sometimes makes it difficult to follow.

– In all the manuscript there is no mention of significance in the violin and box plots when the control KD and the *kipferl* KD are compared. As the manuscript now stands, the reader can have the impression that the changes might be important but non-significant.

– For each UCSC genome browser track, the genomic coordinates of the presented region should be added either on the figure or in the legend.

Figure Sup 3:

– Based on the figure legend, there are some inversions the E is the B, the B◊D, the D◊E.

– The picture of the FISH experiments of the *42AB* and *38C* are not really convincing. Furthermore, it is surprising not to see any *80F* FISH experiments.

– Figure 4D. the name of the TEs and the presence in cluster *42AB, 38C* is very informative. It should also be added for the Kipferl dependent TE piRNAs. The presence in the cluster 80F should also be described.

– The scale of the UCSC genome browser tracks are variable. For a given locus it is important to keep the same scale between different genotypes. Figure 1C, considering the heterochromatic region and the *Cluster 80F*, the Rhino ChIP-seq track is at 300 for the *w1118* and MTD-Gal4 genotypes whereas it is at 150 for the *Iso1* genotype. It is the same problem in Figure 5C for the Kipferl ChIP-seq results on the *42AB* cluster.

---

## [Author Response]

Essential revisions:1 – Kipferl was identified as a Rhino partner by yeast two-hybrid. However, independent confirmation by co-IP experiments seems to be missing. Please provide such experimental validation.

We have attempted this experiment several times using various buffer conditions. A co-IP of Rhino and Kipferl has proven to be very challenging. In our experience, most of Kipferl is tightly associated with chromatin and cannot be solubilized under buffer conditions that are compatible with the Rhino interaction. Moreover, co-expression of the two proteins in S2 cells for pull-down Western experiments did not allow any conclusions, as we observed background binding of Kipferl to the beads. However, we have conducted a complementary experiment that strongly supports that Rhino and Kipferl do not only interact in a heterologous yeast two-hybrid assay, but that the two proteins are also in very tight physical proximity in vivo. Briefly, a TurboID experiment (taking advantage of GFP-tagged Rhino in flies co-expressing the TurboID Biotin ligase fused to a GFP nanobody) recovered the complete set of known Rhino interactors, and in addition CG2678/Kipferl. As a control, we used flies expressing nuclear GFP alone alongside TurboID-GFP_nanobody (in a triplicate experiment coupled to label free quantitative mass spectrometry). We added this important complementary evidence for a Rhino-Kipferl interaction to Figure 2.

2 – Regarding the potential position of Kipferl at the interface of a genetic conflict, reviewers were interested to know whether Kipferl was conserved outside of melanogaster and more particularly, in *simulans*, and if so, whether some rapidly evolving amino-acids could be identified between the two species.

Based on sequence conservation and gene synteny, Kipferl orthologs can be confidently identified in several other Drosophilid species, including *D. simulans*. We added the protein sequence alignment of Kipferl paralogs to Figure 2 Figure supplement 1. As far as we can tell, there are no obvious signs for rapidly evolving amino acids in the region that underlies the Rhino interaction (ZnF#4). We also did not find clear signs of positive selection within Kipferl when analyzing *D. melanogaster* natural strains. The absence of signs for positive selection has been mentioned in the previous Discussion and we slightly expanded that statement (last paragraph). In the future, it will be interesting to perform genetic swap experiments with Kipferl orthologs from other species.

3 – The claim that Kipferl directly binds to DNA should be toned down, as recruitment to Rhino-occupied loci through protein/protein interactions is equally possible at that stage, unless biochemical evidence is provided.

We agree that this is formally possible. We do, however, like to point out that Kipferl does bind chromatin also independently of Rhino (at Kipferl-only loci), foremost based on the data where we expressed Kipferl in the OSC line that does not express detectable Rhino. We toned down the argumentation in the manuscript accordingly throughout the text relating to Figures 5 and 6.

4 – Please integrate the role of maternally deposited piRNAs into the model of Rhino recruitment to piRNA loci, and where they stand regarding Kipferl. For example, are these two modes of recruitment temporally separated (maternal piRNAs being more in stem/progenitor cells of the female germline, while Kipferl may be acting in more differentiated female germ cells, and be therefore involved in maintenance rather than original specification)? Do maternal piRNAs and Kipferl activate different piRNA clusters? Or do they act redundantly? No specific experiments are required here, but the view(s) of the authors on that matter would be very useful to share, as maternal piRNA-dependent specification of piRNA-producing loci is a well-accepted model.

This is indeed an important point. Our data do not allow strong conclusions at this point. We extended this part of the discussion to reflect our current view on this topic (second paragraph).

5 – The authors propose that Kipferl acts upstream from Rhino but the hierarchy between Kipferl and Rhino seems more complicated than this simple model, as Kipferl binding at piRNA loci can also be reduced or even completely gone in Rhino mutants. The authors indeed discuss that Rhino is required for ectopic Kipferl spreading to satellite loci, but it seems that Rhino is also required to some extent for Kipferl to bind at canonical piRNA loci. The authors are not required to carry out a tethering experiment-as suggested by reviewer #2- but the text should be modulated accordingly.

The reviewers are correct in pointing out that our data is not consistent with a strict Kipferl -> Rhino hierarchy. This is as Rhino has a clear impact on Kipferl’s chromatin binding capacity as well. As we do find Kipferl-only loci but not any clear Rhino-only loci, we argue that Kipferl is capable of binding DNA/chromatin in the absence of Rhino, but that Rhino has a strong stabilizing and spreading impact on Kipferl at sites that are within an H3K9-methyl domain. It is, however, important to point out that Rhino is capable of binding chromatin also independently of Kipferl in situations, where Kipferl is experimentally deleted (*kipferl* mutant) or in early stages of oogenesis where Kipferl is barely detectable. We added specific comments to this in the text (Results relating to Figure 5 and paragraph 3 of the Discussion).

Figure revisions:– Please provide statistics in violin and box plots. Currently, the reader may have the impression that changes may be present but non-significant.

We added the mean of the fold change for each violin plot (biological effect size), as well as the statistical significance as calculated via student’s t-test (for ChIP-seq enrichments) or Wilcoxon signed-rank test (for sRNA-seq fold changes).

– Genomic coordinates have to be provided on UCSC genome tracks.

We added the coordinates of each screenshot below the locus name in the various figures.

– Please homogenize your scales in figures of genome tracks, particularly when comparing different genotypes for a given locus (see Figures 1C and 5C for example).

We agree with this overall comment. For Figure 1, however, we think it makes more sense to leave the data display as is and to add a comment in the text saying that “*iso1* has the same pattern, but at some loci shows Rhino ChIP-seq signal at a lower level”. We speculate that this might be due to the fact that the *kipferl* locus in *iso1* differs from the *kipferl* locus in all examined experimental fly strains as it lacks part of the linker between the two zinc finger arrays (this is mentioned in the method section, line 810-816). For Figure 5, we made the y-axis scale consistent between genotypes. We also added panels with a zoom-in to show the remaining Kipferl enrichment pattern in better detail.

Reviewer #1 (Recommendations for the authors):1. Without Kipferl, H3K9me3 seems to mildly (but visibly) increase (Figure 3G) while H3K9me2 seems to mildly (but visibly) decrease (Figure S3G). Is there a statistically significant difference between the two genotypes for K9me2/3? Is there a correlation between change in H3K9me2/3 and change in Rhino binding and what is their correlation coefficient?

The reviewer is right. Our data indicate a small shift from H3K9me2 to me3 upon depletion of Kipferl. However, this was not limited to Rhino-bound genomic tiles. In the absence of more quantitative experiments using spike-in normalization approaches, we refrain from making a strong conclusion from this. Our key argument is that Rhino chromatin occupancy is lost at most sites despite high levels of remaining H3K9-methylation.

2. In Abstract, Intro and Discussion, authors describe the exciting possibility that Rhino enrichment on satDNA in *kipferl-/-* could reflect a conflict. Given that Rhino binds satDNA in wildtype, what about the alternative possibility that it is merely a passive consequence of Rhino having to re-localize to Kipferl-independent loci in *kipferl-/-*? Or is there a disproportionate, higher-than-expected enrichment onto satDNA, with respect to *42AB* and *38C*, upon loss of Kipferl? Is there an increase in *42AB* and *38C* transcripts in *kipferl-/-* by RNA FISH? What makes up the remaining 79% piRNAs when 21% are satDNA piRNAs at line 307?

Other than piRNAs originating from clusters *42AB* or *38C* (whose levels remain unchanged or decrease slightly), piRNAs derived from several Satellites increase strongly (up to 10-fold) although total piRNA levels in *kipferl* mutant ovaries decrease by about 40%. We did not observe increased RNA FISH signal for *42AB* and *38C* in the *kipferl* mutants compared to wildtype, while a strong increase in RNA FISH signal is seen for Satellites. This argues in favor of sequestration of the piRNA cluster machinery to Satellites, rather than just a distribution of free Rhino to remaining loci capable of binding Rhino. The remaining 80% piRNAs in *kipferl* mutants are somatic piRNAs (~20%) and piRNAs derived from Rhino dependent source loci where Kipferl is less important (e..g. *cluster 42AB*, *cluster 38C,* telomeres).

3. It is unclear which sequencing experiment is done with replicates and how many replicates, which should be clearly described. For replicated experiment, is a representative dataset OR the average of replicates shown? In sequencing experiments without replicates, how could one be convinced of the result?

We specified the number of biological replicates in the figure legends throughout the manuscript and provide now additional data (Figure 4 - Figure supplement 1C) for full mutant sRNA-seq data (control, *kipferl -/-* and *rhino -/-*) to complement the missing replicates for the previous analysis. These results confirm our analysis from the RNAi-mediated knockdown samples.

4. In all confocal images, is only one z-slice shown? Other than Figure 3C, Rhino seems to have additional foci besides the pastry domain at the absence of Kipferl (e.g., Figures6D, S3C, S3H). Showing the z-projection for single nurse cells might make statements about Rhino localization more convincing, especially the claim that *kipferl, nxf3* double mutants restore Rhino foci that are absent in *kipferl-/-*.

All confocal images are no more than 5-10 z-sections of 0.2 μm distance and are therefore displaying only a part of the entire nucleus. This is stated now in the methods section (relating to RNA FISH and immunofluorescence stainings). In *kipferl* mutant nurse cells, the large peri-nuclear accumulations (the ‘Kipferl’) is the most prominent, but not the only visible Rhino domain. Regarding the Nxf3 comment: This might have been misleading in the text (now formulated more clearly, text relating to Figure 3, last paragraph): In *nxf3-kipferl* double mutants, no Rhino domains are restored. It is just that the Kipferl-structure is not anymore at the nuclear periphery, but rather internalized.

5. The correlation coefficient with respective p-value should be reported whenever a correlation argument is made throughout the manuscript. e.g., lines 161, 347 and throughout Figures7 and S7.

We changed the wording in lines 161 (now 164) and 347 (now 404-405) and added coefficients throughout the text relating to Figure 7.

6. Text description should not exaggerate the result. e.g., "every" at line 556. Text often says Kipferl is important for most/majority of Rhino domains, could it be supported by actual numbers (# out of total #, either # domain or # 1kb tiles)? Given that most active piRNA loci *42AB* and *38C*, as well as telomeric transposons, are all Kipferl-independent, one wonders if Kipferl really controls most/majority of Rhino domains. Since there are many Rhino-only and Kipferl-only domains/1kb tiles, listing actual numbers helps readers contextualize the relationship between Rhino and Kipferl as well as the prevalence of exceptions.

In our data, Kipferl in fact localizes to every Rhino domain in wildtype ovaries (see Figure 2), even though it does not bind upstream of Rhino at all of these sites. When using binary cutoffs of peak-calling algorithms, 97% of all analyzable Rhino domains (in kb) are dependent on Kipferl. Of these, 65% are found within 5 kb up or downstream of a Rhino-independent Kipferl peak.

Reviewer #2 (Recommendations for the authors):This is a compelling story to establish the function of Kipferl in recruiting Rhino to the piRNA-producing loci in *Drosophila* ovarian genome. I enjoyed reading and considering it in the past few days. I felt that there are some questions still unanswered--particularly how Rhino is recruited to *42AB*. However, given the authors already generated massive amount of data and the complexity of Rhino recruiting mechanisms, I don't think it would be fair to ask for an answer here. Here are a few suggestions:1. I think the authors should comment on the potential homolog of Kipferl in other *Drosophila* species. Rhino appears to be rapidly evolving, even simulans Rhino could not be compatible with the melanogaster piRNA machineries. Does Kipferl homolog exist in simulans. If yes, how different it is with the melanogaster Kipferl, especially the amino acids involving in Rhino and DNA binding. Answering these questions could potentially broaden the impact of this story.

Please see our response to the major point #2 above.

2. Line159, H3K9me3 or H3K9me2?

Here we are indeed talking about H3K9me2. We adjusted the text to make this more clear.

3. I am confused about the message from Line 365-367. Do you mean Rhino peaks towards the periphery?

Within the large piRNA cluster *42AB* we observe a restriction of Rhino toward more central regions of the cluster upon loss of Kipfer, with a reduction of Rhino towards the boundaries of the cluster. It is at these more peripheral sites (where Rhino depends on Kipferl even at *cluster 42AB*) that we observe small Kipferl enrichment peaks in the *rhino* mutant Kipferl ChIP-seq data set. These low, but detectable Kipferl binding sites likely contribute to the stabilization of Rhino on these more “peripheral” regions of the cluster. We adjusted the text to convey this better (lines 427-436).

4. Line 452: motives or motifs.

All instances were changed to motifs.

5. The last paragraph is way too long. It is good to only deliver one message per paragraph.

We split this paragraph into two.

Reviewer #3 (Recommendations for the authors):The manuscript, although well written, is very dense with a lot of figures which sometimes makes it difficult to follow.– In all the manuscript there is no mention of significance in the violin and box plots when the control KD and the *kipferl* KD are compared. As the manuscript now stands, the reader can have the impression that the changes might be important but non-significant.

We added statistical significance parameters to the relevant plots as well as mean effect sizes.

– For each UCSC genome browser track, the genomic coordinates of the presented region should be added either on the figure or in the legend.

Coordinates have been added.

Figure Sup 3:– Based on the figure legend, there are some inversions the E is the B, the B◊D, the D◊E.

This has been corrected.

– The picture of the FISH experiments of the *42AB* and *38C* are not really convincing. Furthermore, it is surprising not to see any 80F FISH experiments.

The piRNA cluster FISH experiments aim to establish that the large Rhino accumulations observed in the absence of Kipferl do not correspond to the remaining Rhino-bound piRNA cluster loci. To that end the presented images were selected to depict the large Rhino-GFP domain in the same z plane with foci for both clusters. Due to photo bleaching effects we are not able to image deeper stacks, and likely miss weak Rhino signal that would be expected at 38C and *42AB* foci. We believe that the low levels of Rhino that remain at *42AB* and especially at *38C* are another reason why we don’t see stronger overlap. As *cluster 80F* lacks Rhino occupancy as well as piRNAs, we did not see the importance of a dedicated *cluster 80F* FISH experiment.

– Figure 4D. the name of the TEs and the presence in cluster *42AB, 38C* is very informative. It should also be added for the Kipferl dependent TE piRNAs. The presence in the cluster *80F* should also be described.

We added the information about *cluster 80F* to the existing panel, as well as the cluster information about the top 20 elements losing antisense piRNA counts in the *kipferl* knock down.

– The scale of the UCSC genome browser tracks are variable. For a given locus it is important to keep the same scale between different genotypes. Figure 1C, considering the heterochromatic region and the Cluster *80F,* the Rhino ChIP-seq track is at 300 for the *w1118* and MTD-Gal4 genotypes whereas it is at 150 for the *iso1* genotype. It is the same problem in Figure 5C for the Kipferl ChIP-seq results on the *42AB* cluster.

We changed this. Please also see our comment above (main revision requirements).